# Should Decision-Makers Reveal Classifiers in Online Strategic Classification?

**Han Shao** [1]   **Shuo Xie** [2]   **Kunhe Yang** [3]

## Abstract

Strategic classification addresses a learning problem where a decision-maker implements a classifier over agents who may manipulate their features in order to receive favorable predictions. In the standard model of online strategic classification, in each round, the decision-maker implements and publicly reveals a classifier, after which agents perfectly best respond based on this knowledge. However, in practice, whether to disclose the classifier is often debated—some decision-makers believe that hiding the classifier can prevent misclassification errors caused by manipulation. In this paper, we formally examine how limiting the agents' access to the current classifier affects the decision-maker's performance. Specifically, we consider an extended online strategic classification setting where agents lack direct knowledge about the current classifier and instead manipulate based on a weighted average of historically implemented classifiers. Our main result shows that in this setting, the decision-maker incurs $(1 - \gamma)^{-1}$ or $k_{\text{in}}$ times more mistakes compared to the full-knowledge setting, where $k_{\text{in}}$ is the maximum in-degree of the manipulation graph (representing how many distinct feature vectors can be manipulated to appear as a single one), and $\gamma$ is the discount factor indicating agents' memory of past classifiers. Our results demonstrate how withholding access to the classifier can backfire and degrade the decision-maker's performance in online strategic classification.

## 1. Introduction

In online strategic classification, a decision-maker makes decisions over a sequence of agents who may manipulate

[1]CMSA, Harvard University [2]Toyota Technological Institute at Chicago [3]University of California, Berkeley. Correspondence to: Han Shao <han@cmsa.fas.harvard.edu>, Shuo Xie <shuox@ttic.edu>, Kunhe Yang <kunheyang@berkeley.edu>.

*Proceedings of the $42^{nd}$ International Conference on Machine Learning*, Vancouver, Canada. PMLR 267, 2025. Copyright 2025 by the author(s).

their features to receive favorable outcomes (Brückner & Scheffer, 2011; Hardt et al., 2016; Ahmadi et al., 2023). For example, in college admissions, when a decision-maker evaluates applicants, students may retake the SAT, switch schools, or enroll in easier classes to boost their GPAs in hopes of gaining admission. Similarly, in loan approval, where a classifier assesses applicants based on their credit scores, individuals may open or close credit cards or bank accounts to improve their credit scores.

More formally, an agent $(x, y)$ consists of a pre-manipulation feature vector $x$ and a ground-truth label $y$. When modeling strategic manipulation, we assume that each agent has limited manipulation power. We adopt manipulation graphs, initially introduced by Zhang & Conitzer (2021), to represent feasible manipulations. In this graph, the nodes correspond to all possible feature vectors. There is an edge from $x$ to $x'$ if and only if an agent with the feature vector $x$ can manipulate their feature vector to $x'$. Additionally, there is always a self-loop at each node, as agents can choose to remain at $x$ by doing nothing.

A sequence of agents $(x_1, y_1), (x_2, y_2), \ldots$ arrives sequentially. In each round $t$, the decision-maker implements a classifier $h_t$, and the agent $(x_t, y_t)$ manipulates their feature vector from $x_t$ to a neighbor $v_t$ in an attempt to receive a positive prediction under $h_t$. In prior work, it is usually assumed that the decision-maker is transparent—they reveal their current classifier to the agents before manipulation—and that agents are rational—they best respond by manipulating to a reachable feature that is labeled as positive (if one exists) in the most cost-efficient way. Specifically, if all their neighbors are labeled as negative, they remain at their current feature without making any changes, as manipulation would not help change the prediction from negative to positive. Many algorithms developed in prior work (e.g. Ahmadi et al., 2023; 2024; Cohen et al., 2024a) heavily rely on this standard tie-breaking assumption.

However, in real-world applications, the decision-maker may withhold their classifier to prevent agents from manipulating their features. Nevertheless, even without knowledge of the current classifier, agents will still attempt to optimize their outcomes through strategic manipulation. This raises a fundamental question:

**Can decision-makers benefit from hiding the classifiers?**

More specifically, we assume that agents only have knowledge of previously implemented classifiers when they respond. For example, in centralized college admissions, universities often publish past cutoff scores, revealing previous classifiers.

In this setting, we consider natural models of agents' behavior that exhibit heuristic rationality. One natural approach is for agents to best respond to the classifier from the previous round. For example, student applicants may assume that the admission rule will closely resemble the one used last year. Another heuristic method is to best respond to the average of historically implemented classifiers. We unify these approaches by considering a general framework where agents best respond to a weighted average of historical classifiers.

Under this model, we answer our central question negatively. We show that when agents best respond to a weighted average of past classifiers, the decision-maker incurs $(1-\gamma)^{-1}$ or $k_{\text{in}}$ times more mistakes, where $k_{\text{in}}$ is the maximum indegree of the manipulation graph and $\gamma \in (0,1)$ is the discount factor representing agents' memory of past classifiers. This implies that the decision-maker may not benefit from hiding the classifiers but incur a substantial increase in the mistake bound.

To better understand this problem, we first analyze a simple base case by removing the standard tie-breaking assumption—that agents remain at their original feature vector $x$ when their entire neighborhood is labeled as negative. We show that removing this assumption can already hurt the decision-maker, increasing the mistake bound by a factor of $\Theta(k_{\text{in}})$. This not only serves as a building block but also highlights that even a small perturbation in agents' perceived classifiers can significantly impact the mistakes made by the decision-maker.

**Contributions.** We summarize our contributions as follows.

- When agents best respond to the current classifier with arbitrary tie-breaking, we provide a lower bound on the mistake bound, showing that the decision-maker will make $\Omega(k_{\text{in}})$ times more mistakes. We also propose an algorithm that achieves a matching upper bound.
- When agents best respond to the weighted average of history with arbitrary tie-breaking, we establish a lower bound on the mistake bound, showing that the decision-maker will make $(1-\gamma)^{-1}$ or $k_{\text{in}}$ times more mistakes. Additionally, we propose an algorithm with mistake bound of $O((1-\gamma)^{-1} \cdot k_{\text{in}})$.
- We also explore the scenario where agents run online learning algorithms, and discuss the challenge of modeling learning agents through diminishing regret due to their nonstatic and large action space.

## 1.1. Related Work

The research on strategic machine learning—which focuses on designing machine learning algorithms that remain robust in the presence of strategic behaviors—dates back to the works of Dalvi et al. (2004); Brückner & Scheffer (2011); Dekel et al. (2010). Within this field, *strategic classification* was first studied by Hardt et al. (2016), who examines the accuracy of classification tasks when agents can modify their features to receive more favorable outcomes. Overall, our work contributes to the literature of strategic classification in the following three aspects.

**(I) Learnability in the online setting.** Our work focuses on the *online* setting where the decision-maker (also called the *learner*) aims to make irrevocable classification decisions to a sequence of agents arriving one by one. Among the prior works that analyze the mistake bound/Stackelberg regret in such settings (Dong et al., 2018; Chen et al., 2020; Ahmadi et al., 2021; 2023; 2024; Shao et al., 2023; Cohen et al., 2024a), our work is most closely related to that of Ahmadi et al. (2023; 2024); Cohen et al. (2024a). These works model the agents' ground-truth labels as induced by graph-based manipulations (Zhang & Conitzer, 2021; Lechner & Urner, 2022) toward an unknown hypothesis within a given hypothesis class with bounded complexity, an assumption analogous to *realizability* in classical online learning (Littlestone, 1988). Building on this framework, our work is the first to analyze how the mistake bound changes under alternative models of agent behaviors, specifically when agents best respond to a weighted sum of historical classifiers rather than the current classifier.

**(II) Agent behavioral models beyond best response.** There have also been recent works exploring alternative agent behaviors beyond the simple best-response model, including noisy response (Jagadeesan et al., 2021), gradient descent (Zrnic et al., 2021), and non-myopic agents optimizing for discounted future rewards (Haghtalab et al., 2022). In contrast, our settings assume that agents cannot observe the current classifier but manipulate based on historical observations. Xie & Zhang (2024) also assumes agents best respond to previous decision policy but they focus on welfare and fairness. We compare the learner's optimal mistake bound both when the classifier is revealed and when it is withheld. More broadly, these studies fit within the context of repeated Stackelberg (principal-agent) games with learning agents, (e.g. Haghtalab et al., 2022; 2024; Blum et al., 2014), where recent work focuses on designing principal's algorithms to strategize against agents who employ specific online learning algorithms such as no-regret algorithms (Braverman et al., 2018; Deng et al., 2019; Mansour et al., 2022; Fiez et al., 2020; Brown et al., 2024). While these works generally assume known or fixed/stochastic games, part of our main challenge arises from the fact that agents' initial fea-

tures are unknown and chosen by an online adversary. Furthermore, as we will discuss in Section 5, defining learning agents in strategic classification presents unique challenges due to the dynamic and large action spaces.

**(III) Decision-maker's strategy beyond full transparency.** Our study studies the decision-maker's choice between making the classifiers fully transparent or withholding information. Prior works have explored other forms of partial transparency mechanisms in the *offline* setting, such as releasing a subset of potential classifiers that includes the actual implemented classifier (Cohen et al., 2024b), incorporating randomness and noise (Braverman & Garg, 2020), withholding the classifier so that agents employ imitation and social learning (Ghalme et al., 2021; Bechavod et al., 2022; Akyol et al., 2016), providing feedback on the classifier with varying levels of accuracy (Barsotti et al., 2022), and revealing counterfactual explanations instead of the classifier itself (Tsirtsis & Gomez Rodriguez, 2020). Our main contribution to this thread is studying the *online* interaction between the decision-maker and a stream of agents, and to quantitatively characterize the impact of information disclosure strategies on the decision-maker's performance.

While our work also focuses on the accuracy perspective, many works have explored various other perspectives of strategic classification, such as encouraging genuine improvements versus discouraging "gaming" (e.g. Ahmadi et al., 2022; Bechavod et al., 2021; Haghtalab et al., 2021; Liu et al., 2020), understanding the causal implications (e.g. Miller et al., 2020; Bechavod et al., 2021; Shavit et al., 2020; Perdomo et al., 2020), and fairness concerns (e.g. Hu et al., 2019; Liu et al., 2020; Milli et al., 2019).

## 2. Model

Throughout this work, we focus on the binary classification task in the online setting. Let $\mathcal{X}$ denote the feature vector space and $\mathcal{Y} = \{0, 1\}$ denote the label space. A hypothesis $h : \mathcal{X} \mapsto \mathcal{Y}$ (also called a classifier or a predictor) is a function that maps feature vectors to labels. We denote by $\mathcal{H} \subset \mathcal{Y}^{\mathcal{X}}$ the hypothesis class. A fractional predictor $f : \mathcal{X} \mapsto [0, 1]$ is a function that maps feature vectors to the probability of being labeled as 1.

An example $(x, y) \in \mathcal{X} \times \mathcal{Y}$, referred to as an *agent* in this context, consists of a pre-manipulation feature vector $x$ and a ground-truth label $y$. We consider the task of sequential classification where the decision-maker (aka learner) aims to classify a sequence of agents $(x_1, y_1), (x_2, y_2), \ldots$ that arrives in an online manner. In each round $t$, the decision-maker implements a classifier $h_t$, and the agent $(x_t, y_t)$ manipulates their feature vector from $x_t$ to $v_t$ with an attempt to receive a positive prediction under $h_t$. The interaction between the decision-maker and the agents (which repeats

for $t = 1, \ldots, T$) is described in Protocol 1.

---
**Protocol 1** Decision-Maker/Agent Interaction
---
1: **for** $t = 1, \ldots, T$ **do**
2:     The learner implements a classifier $h_t \in \mathcal{Y}^{\mathcal{X}}$.
3:     The environment picks an agent $(x_t, y_t)$.
4:     The agent manipulates from $x_t$ to $v_t$.
5:     The learner observes the post-manipulation feature vector $v_t$ and the true label $y_t$, and makes a mistake if $h_t(v_t) \neq y_t$.
6: **end for**

---

**Manipulation Graph.** Agents will try to maximize their chance of receiving a positive prediction by modifying their feature vector to some reachable feature vectors. Following prior works (e.g. Zhang & Conitzer, 2021; Lechner & Urner, 2022; Ahmadi et al., 2023; Lechner et al., 2023), we model the set of reachable feature vectors through a manipulation graph $G = (\mathcal{X}, \mathcal{E})$, where the nodes are all feature vectors in $\mathcal{X}$. For any two feature vectors $x, x' \in \mathcal{X}$, there is a directed edge from $x$ to $x'$—i.e., $(x, x') \in \mathcal{E}$—if and only if an agent with initial feature vector $x$ can manipulate to $x'$. Additionally, there is always a self-loop at each node $x \in \mathcal{X}$, i.e., $\forall(x, x) \in \mathcal{E}$, as agents can choose to remain at $x$ by doing nothing.

For each node $x \in \mathcal{X}$, let $N_{\text{out}}[x] = \{x'|(x, x') \in \mathcal{E}\}$ and $N_{\text{in}}[x] = \{x'|(x', x) \in \mathcal{E}\}$ denote the out- and in-neighborhoods of $x$ in the manipulation graph $G$. Let $k_{\text{out}} = \sup_{x \in \mathcal{X}} |N_{\text{out}}[x]|$ and $k_{\text{in}} = \sup_{x \in \mathcal{X}} |N_{\text{in}}[x]|$ denote the maximum out-degree and maximum in-degree of the manipulation graph, respectively.

**Agent Behavior Models.** At each round $t$, an agent manipulates their feature vector from $x_t$ to a reachable vector $v_t \in N_{\text{out}}[x_t]$ with the intent to maximize their probability of receiving a positive prediction. Concretely, the agent either observes $h_t$ or forms an estimate $\widetilde{h}_t$ based on the information available to them, and then chooses $v_t \in N_{\text{out}}[x_t]$ among those that maximize $h_t(v)$ or $\widetilde{h}_t(v)$. We formalize the set of such best response manipulations as the *best response set*:

**Definition 2.1** (Best Response Set). Given a (potentially fractional) predictor $h \in [0, 1]^{\mathcal{X}}$, the best response function $\text{BR}_h : \mathcal{X} \mapsto 2^{\mathcal{X}}$ will map $x$ to the set of reachable neighbors with the highest predicted value, i.e.,

$$\text{BR}_h(x) := \text{argmax}_{v \in N_{\text{out}}[x]} h(v).$$

We consider three models of agent behaviors based on (1) how they break ties when there are multiple options inside $\text{BR}_h(x)$, and (2) how agents form their estimate $\widetilde{h}_t$.

- **(Revealed-Std) Revealed Classifier + Best-Response with Standard Tie-breaking.** This is the standard setting

in the literature of strategic classification, e.g., Ahmadi et al. (2023); Cohen et al. (2024a); Ahmadi et al. (2024). In this case, the decision-maker reveals the implemented 0-1 classifier $h_t \in \{0,1\}^{\mathcal{X}}$ to the agent in each round. The agent best responds to $h_t$ by choosing $v_t \in \mathsf{BR}_{h_t}(x_t)$. Specifically, if every reachable feature vector in $N_{\text{out}}[x_t]$ is labeled as 0 under $h_t$—i.e., when $h_t(\mathsf{BR}_{h_t}(x_t)) = 0$—the agent chooses $v_t = x_t$ as they have no incentive to manipulate if manipulation does not yield a positive classification. Otherwise, if there are neighbors labeled as 1, ties among them are broken arbitrarily.

- (Revealed-Arb) **Revealed Classifier + Best-Response with Arbitrary Tie-breaking.** As a variation of Revealed-Std, in Revealed-Arb, the agents still observe $h_t$ and best responds to it, but break ties arbitrarily among the vectors in $\mathsf{BR}_{h_t}(x_t)$[1]. In particular, even if all neighbors in $N_{\text{out}}[x_t]$ are labeled negative, the agent may still choose to move rather than stay, due to the arbitrary nature of tie-breaking. We introduce this model as an intermediary step toward our main setting, where agents do not observe $h_t$ directly, but instead best respond to an estimate $\widetilde{h}_t$. Estimation errors can make some out-neighbors appear more favorable — even when all are actually negative under $h_t$ — so agent's best response to $\widetilde{h}_t$ may resemble a best response to $h_t$ under arbitrary tie-breaking. This motivates us to consider the Revealed-Arb setting.

- ($\gamma$-Weighted) **Unrevealed Classifier Estimated via Weighted Sum of History.** This is the main setting of this paper. In each round $t$, the agent cannot observe the implemented classifier $h_t$, but only has access to the historically implemented classifiers $h_1, \ldots, h_{t-1}$. As a natural heuristic, the agent estimates $h_t$ by taking a weighted average of these historical classifiers,

$$\widetilde{h}_t^{\gamma} = \frac{\sum_{\tau=0}^{t-2} \gamma^{\tau} h_{t-1-\tau}}{\sum_{\tau=0}^{t-2} \gamma^{\tau}} = \frac{1-\gamma}{1-\gamma^{t-1}} \cdot \sum_{\tau=0}^{t-2} \gamma^{\tau} h_{t-1-\tau},$$

$\gamma \in (0,1)$ is a discount factor that reflects how quickly the agent's memory of older classifiers diminishes. The agent then chooses $v_t \in \mathsf{BR}_{\widetilde{h}_t^{\gamma}}(x_t)$ with *arbitrary tie-breaking*. By adjusting the values of $\gamma$, this model captures a spectrum of agent behaviors. When $\gamma \to 1$, $\widetilde{h}_t^{\gamma}$ approaches the average of all historical classifiers, reminiscent of *fictitious play* in repeated games. When $\gamma \to 0$, $\widetilde{h}_t^{\gamma}$ approaches $h_{t-1}$, meaning that the agent uses a one-step memory and best responds only to the most recent classifier.

---

[1] The tie-breaking may be adversarially chosen to induce the maximum number of mistakes. However, this differs from adversarial perturbations studied in robustness literature (e.g., (Montasser et al., 2019)), since the agent only chooses from vectors that maximize $h_t$.

**Learner's Objective.** The learner's objective is to minimize the total number of classification mistakes. For a sequence of agents $S = (x_t, y_t)_{t \in [T]}$, the learner's mistake bound is

$$\mathcal{M}_{\phi}(S) = \sum_{t=1}^{T} \mathbb{1}[h_t(v_t) \neq y_t],$$

where $\phi \in \{\text{Revealed-Std}, \text{Revealed-Arb}, \gamma\text{-Weighted}\}$ denotes one of the three agent behavioral models described above, which determines how $v_t$ is chosen. We also use $\mathcal{M}_{\phi}(\mathcal{A}, S)$ to denote the mistake bound of a specific algorithm $\mathcal{A}$. Following prior works (Ahmadi et al., 2021; 2023; Cohen et al., 2024a), we focus on the *strategic realizable* setting, where there exists an optimal hypothesis $h^{\star} \in \mathcal{H}$ that perfectly classifies the agent sequence $S$ when each agent $(x_t, y_t)$ best responds to $h^{\star}$. Formally, this means

$$\forall (x_t, y_t) \in S, \quad h^{\star}(\mathsf{BR}_{h^{\star}}(x_t)) = y_t.$$

We focus on this realizable setting because, across all three agent behavior models that we consider, there exists a learner strategy that achieves zero mistakes by using the optimal-in-hindsight classifier $h^{\star}$ across all rounds. Consequently, the mistake bound $\mathcal{M}(S)$ aligns with the notion of *Stackelberg regret*, which measures the learner's performance relative to the best fixed strategy in hindsight in this principal-agent framework. In either of the three settings $\phi \in \{\text{Revealed-Std}, \text{Revealed-Arb}, \gamma\text{-Weighted}\}$, we use

$$\mathcal{M}_{\phi}(\mathcal{H}) = \sup_{S \text{ strategic realizable under } \mathcal{H}} \mathcal{M}_{\phi}(S)$$

to denote a learner's worst-case mistake bound across all strategic realizable sequences. Similar to $\mathcal{M}_{\phi}(\mathcal{A}, S)$, we sometimes use $\mathcal{M}_{\phi}(\mathcal{A}, \mathcal{H})$ to emphasize the algorithm $\mathcal{A}$. Following a common assumption in the literature (Cohen et al., 2024a; Ahmadi et al., 2024; 2023), we focus on deterministic algorithms.

In the standard setting, Ahmadi et al. (2023); Cohen et al. (2024a) show that $\mathcal{M}_{\text{Revealed-Std}}(\mathcal{H}) = O(k_{\text{out}} \cdot \text{Ldim}(\mathcal{H}))$, where $\text{Ldim}(\mathcal{H})$ is the Littlestone dimension of $\mathcal{H}$ (Littlestone, 1988). They also establish that $\Omega(k_{\text{out}} \cdot \text{Ldim}(\mathcal{H}))$ is a tight lower bound for all deterministic algorithms. In this paper, we characterize the optimal mistake bound in the other two settings of Revealed-Arb and $\gamma$-Weighted.

### 2.1. Summary of Results

- We show that in the Revealed-Arb setting, the learner suffers an extra $k_{\text{in}}$ multiplicative factor in the mistake bound than the Revealed-Std setting. More specifically, we show that there exists family of instances where $\Omega(k_{\text{in}} k_{\text{out}} \cdot \text{Ldim}(\mathcal{H}))$ is a lower bound for $\mathcal{M}_{\text{Revealed-Arb}}(\mathcal{H})$ for determinisitc algorithms (Thm 3.3). We also propose an algorithm that reduces our learning problem to classical online learning problem and prove that $\mathcal{M}_{\text{Revealed-Arb}}(\mathcal{H}) = \widetilde{O}(k_{\text{in}} k_{\text{out}} \cdot \text{Ldim}(\mathcal{H}))$, see Cor 3.2.

- When agents lack information of the current classifier, and instead best respond to $\widetilde{h}_t^\gamma$ with a discount factor $\gamma \in (0, 1)$, we propose a reduction algorithm that transforms the $\gamma$-Weighted setting into the Revealed-Arb setting, achieving

$$\mathcal{M}_{\gamma\text{-Weighted}}(\mathcal{H}) = O\big((1-\gamma)^{-1}\big) \cdot \mathcal{M}_{\text{Revealed-Arb}}(\mathcal{H})$$
$$= \widetilde{O}\left((1-\gamma)^{-1} \cdot k_{\text{in}} k_{\text{out}} \cdot \text{Ldim}(\mathcal{H})\right).$$

This implies a $\widetilde{O}\left((1-\gamma)^{-1} \cdot k_{\text{in}}\right)$ multiplicative gap compared to the Revealed-Std deterministic lower bound of $\Omega(k_{\text{out}} \cdot \text{Ldim}(\mathcal{H}))$ (Ahmadi et al., 2023; Cohen et al., 2024a). We also show in Theorem 4.4 that for deterministic algorithms, both the $(1-\gamma)^{-1}$ and $k_{\text{in}}$ factors are necessary in different regimes of the $\gamma$-Weighted setting.

## 3. Revealed Classifiers with Arbitrary Tie-breaking

In this section, we study Revealed-Arb setting, which serves as a building block for our main setting where the classifiers are not revealed. In the Revealed-Arb setting, the decision-maker reveals the classifier $h_t$ at each round, then the agent $(x_t, y_t)$ manipulates their feature vector from $x_t$ to $v_t$ that is chosen *arbitrarily* from the best response set, i.e.,

$$v_t \in \text{BR}_{h_t}(x_t).$$

In particular, this model allows agents to move arbitrarily to any out-neighbor, even when both $v_t$ and $x_t$ are labeled negative by $h_t$. In contrast, prior works such as Ahmadi et al. (2023); Cohen et al. (2024a); Ahmadi et al. (2024) assumes that agents remain at $x_t$ when $h_t$ labels all the out-neighbors $N_{\text{out}}[x_t]$ as negative. This assumption is critical to their results as it allows the decision-maker to exactly infer an agent's true feature vector when it ends up labeled as negative by the classifier. Specifically, when $h_t(v_t) = 0$, the decision-maker can deduce that $x_t = v_t$ and penalize all the hypotheses in $\mathcal{H}$ that misclassify this agent.

**Understanding arbitrary tie-breaking as an intermediate step toward the $\gamma$-Weighted setting.** Addressing the challenges arising from arbitrary tie-breaking is an important intermediate step towards our main $\gamma$-Weighted setting. In the $\gamma$-Weighted setting, agents lack access to the current classifier and instead use the weighted sum of historically implemented classifiers as an estimate. This estimation inevitably introduces small errors that cause the agents to perceive certain out-neighbors as more favorable than others, even when all are labeled as negative under $h_t$. As a result, when the agent's estimate $\widetilde{h}_t$ is close to $h_t$, their best response to $\widetilde{h}_t$ can still be viewed as a best response to $h_t$, but under arbitrary tie-breaking rules (which we will formalize in Section 4). Therefore, we first remove the standard tie-breaking assumption, and study the Revealed-Arb setting before analyzing the $\gamma$-Weighted setting.

We will present our upper bound in the Revealed-Arb setting in Section 3.1 and lower bound in Section 3.2.

### 3.1. Upper Bound

In this section, we provide an upper bound on $\mathcal{M}_{\text{Revealed-Arb}}$ by establishing a reduction to classical (non-strategic) online learning. While the main idea is similar to that of Cohen et al. (2024a), we need to address the challenge of no longer being able to observe the agent's true feature vector $x_t$ under false negative mistake (i.e., when the agent's true label is $y_t = 1$, but the classifier $h_t$ incorrectly labels the agent as 0). We address this challenge via a more careful and conservative procedure of updating the expert class. We present the algorithm in Algorithm 1.

At a high level, Algorithm 1 maintains a set $E$ of weighted experts and predicts $x$ as 1 only when the total weight of experts predicting it as 1 is sufficiently large. Each expert runs a classical non-strategic online learning algorithm, such as the SOA algorithm (Littlestone, 1988), with different inputs. Each time a mistake is made, the algorithm penalizes the experts who made incorrect predictions. The challenge lies in identifying whether an expert has made a mistake, as we do not know the original feature vector $x_t$ and thus cannot determine the post-manipulation feature vector had we followed this expert. Additionally, we aim for at least one expert to perform nearly as well as the optimal classifier $h^\star$. To achieve this, we seek to feed the expert the post-manipulation feature vector that would have resulted had we implemented the optimal classifier $h^\star$.

The algorithm uses the observed $v_t \in \text{BR}_{h_t}(x_t)$ to enumerate all possible manipulated feature vectors that would have allowed the agent to be correctly classified had they best responded to the optimal classifier $h^\star$. Then, at least one expert in $E$ is fed with the same trajectory induced by best-responding to $h^\star$.

In particular, when Algorithm 1 incurs a *false negative* mistake, the agent might have manipulated from an in-neighbor $x_t \in N_{\text{in}}[v_t]$, as permitted by the adversarial tie-breaking rule. To account for this, our algorithm first identifies all the in-neighbors of $v_t$ that cannot manipulate to receive a positive classification under $h_t$—i.e., those who has $h_t(N_{\text{out}}[x]) = 0$—as possible candidates for $x_t$. This forms the set $\widetilde{\mathcal{X}}_t$ as described in Line 9 of Algorithm 1. Our algorithm then considers all potential manipulations of $x_t$ under $h^\star$ and forms the set $\widetilde{\mathcal{V}}_t$ in Line 10. These nodes are then used to update the experts as a way of enumerating all possible trajectories consistent with $h^\star$.

We provide the guarantee for Algorithm 1 in Theorem 3.1 and defer its proof to Appendix A.1.

**Theorem 3.1** (Upper Bound on $\mathcal{M}_{\text{Revealed-Arb}}$)**.** *Let $M^{ns}(\mathcal{A}, \mathcal{H})$ be the mistake bound of $\mathcal{A}$ in the standard (non-*

**Algorithm 1** Reduction from Revealed-Arb to classical (non-strategic) online learning

---

1: **Input:** A standard online learning algorithm $\mathcal{A}$, manipulation graph $G$, maximum out-degree $k_{\text{out}}$, and maximum in-degree $k_{\text{in}}$
2: **Initialization:** Expert set $E = \{\mathcal{A}\}$. Initial weight $w_{\mathcal{A}} = 1$.
3: **for** $t = 1, 2, \ldots$ **do**
4:    Prediction: $\forall x \in \mathcal{X}$, set $h_t(x) = 1$ if and only if

$$\sum_{A \in E : A(x)=1} w_A \geq \frac{\sum_{A \in E} w_A}{2(k_{\text{out}} + 1)(k_{\text{in}} + 1)}.$$

5:    Update: *//when $h_t$ makes a mistake at the observed node $v_t$*
6:    **if** $h_t(v_t) = 1$ and $y_t = 0$ (false positive mistake) **then**
7:       for all $A \in E$ satisfying $A(v_t) = 1$, update $A$ by feeding it with $(v_t, 0)$, update weight $w_A \leftarrow \frac{1}{2} w_A$
8:    **else if** $h_t(v_t) = 0$ and $y_t = 1$ (false negative mistake) **then**
9:       $\widetilde{\mathcal{X}}_t \leftarrow \{x \in N_{\text{in}}[v_t] \mid h_t(N_{\text{out}}[x]) = 0\}$ *//Possible candidates for $x_t$ based on observed $v_t$*
10:      $\widetilde{\mathcal{V}}_t \leftarrow \cup_{x \in \widetilde{\mathcal{X}}_t} N_{\text{out}}[x]$ *//Possible manipulations of $x_t$ under the optimal classifier $h^\star$*
11:      **for all** $A \in E$ s.t. $\forall x' \in \widetilde{\mathcal{V}}_t, A(x') = 0$, **do**
12:         feed $(x', 1)$ to $A$ for each $x' \in \widetilde{\mathcal{V}}_t$ to obtain a new expert $A(x', 1)$
13:         Replace $A$ in $E$ with $\{A(x', 1) \mid x' \in \widetilde{\mathcal{V}}_t\}$, assign weights $w_{A(x',1)} = w_A / (2|\widetilde{\mathcal{V}}_t|)$
14:      **end for**
15:    **end if**
16: **end for**

---

strategic) online learning setting when the inputs are (non-strategic) realizable by $\mathcal{H}$. In online strategic classification where the agent sequence is strategic realizable under $\mathcal{H}$, and each agent best responds to $h_t$ with adversarial tie-breaking, Algorithm 1 achieves

$$\mathcal{M}_{\text{Revealed-Arb}}(\mathcal{H}) = O(k_{in} \cdot k_{out} \cdot \ln(k_{in}k_{out})) \cdot M^{ns}(\mathcal{A}, \mathcal{H}).$$

It is well-known that the Standard Optimal Algorithm (SOA) (Littlestone, 1988) achieves mistake bound $M^{ns}_{\text{SOA}}(\mathcal{H}) = \text{Ldim}(\mathcal{H})$ in the standard realizable online learning (where $\text{Ldim}(\mathcal{H})$ is the Littlestone dimension of $\mathcal{H}$). By plugging this non-strategic mistake bound into Theorem 3.1, we obtain an upper bound on $\mathcal{M}_{\text{Revealed-Arb}}$, summarized in the following corollary.

**Corollary 3.2** (Upper bound on $\mathcal{M}_{\text{Revealed-Arb}}$). *For any hypothesis class $\mathcal{H}$ with Littlestone dimension $d$, and any manipulation graph $G$ with out-degree $k_{out}$ and in-degree*

$k_{in}$*, in the strategic realizable setting, when the agents best-respond to $h_t$ with adversarial tie-breaking, our Algorithm 1 with $\mathcal{A} = SOA$ can achieve mistake bound*

$$\mathcal{M}_{\text{Revealed-Arb}}(\mathcal{H}) = O(k_{in} \cdot k_{out} \cdot \ln(k_{in}k_{out}) \cdot d).$$

### 3.2. Lower Bound

In this section, we provide a lower bound of $\mathcal{M}_{\text{Revealed-Arb}}$ that matches the upper bound up to logarithm factors. We defer the proof of Theorem 3.3 to Appendix A.2.

**Theorem 3.3** (Lower bound on $\mathcal{M}_{\text{Revealed-Arb}}$). *For any $d, k_{in}, k_{out} \in \mathbb{N}$ with $k_{out} \geq k_{in}$, there exists a hypothesis class $\mathcal{H}$ with Littlestone dimension $d$, and manipulation graph $G$ with maximum in (resp. out)-degree $k_{in}$ (resp. $k_{out}$), such that in the strategic realizable setting, any deterministic learning algorithms must suffer*

$$\mathcal{M}_{\text{Revealed-Arb}}(\mathcal{H}) = \Omega(k_{in} \cdot k_{out} \cdot d).$$

Our construction is built upon the lower bounds in Ahmadi et al. (2023); Cohen et al. (2024a). We defer the formal construction to Appendix A.2. Here, we provide some intuition on why removing the standard tie-breaking assumption makes the problem harder.

Consider a manipulation graph that has a subgraph structure shown in Figure 1. Suppose we have a hypothesis class where each hypothesis $h_i$ labels agents with the original feature vector $x_i$ ($\forall i = 1, 2, 3$) as positive and all others as negative. Let the optimal hypothesis be $h_{i^\star}$.

Consider the scenario where the decision-maker implements the all-negative function and the adversary selects agent $x_{i^\star}$. In the Revealed-Arb setting, if the tie-breaking always favors $x_0$, the decision-maker makes a mistake but observes the post-manipulation feature at $x_0$, thereby gaining no information about $i^\star$. This is in contrast to the Revealed-Std setting, where agent $x_{i^\star}$ cannot obtain a positive prediction through manipulation and thus remains at $x_{i^\star}$, enabling the learner to successfully identify the optimal hypothesis.

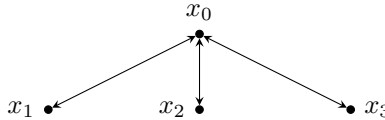

*Figure 1.* A subgraph of the lower bound construction.

## 4. Best Response to the Weighted-Sum of History

In this section, we study the $\gamma$-Weighted setting. In round $t$, the agents cannot observe the current implemented classifier $h_t$. They will only best respond to the weighted average

of historical classifiers $\widetilde{h}_t^\gamma = \frac{1-\gamma}{1-\gamma^{t-1}} \cdot \sum_{\tau=0}^{t-2} \gamma^\tau h_{t-1-\tau}$ for some $\gamma \in (0,1)$ as defined in Section 2, i.e., $v_t \in \mathsf{BR}_{\widetilde{h}_t^\gamma}(x_t)$ with adversarially tie-breaking. We show that the learner will suffer $O((1-\gamma)^{-1} \cdot k_{\text{in}})$ times more mistakes.

## 4.1. Upper Bound

**Theorem 4.1** (Upper bound on $\mathcal{M}_{\gamma\text{-Weighted}}$)**.** *For any hypothesis class $\mathcal{H}$ with Littlestone dimension $d$ and any manipulation graph $G$ with out-degree $k_{out}$ and in-degree $k_{in}$, in the strategic realizable setting, when the agents best-respond to the $\gamma$-weighted sum of the history $\widetilde{h}_t^\gamma$, there exists a learning algorithm that achieves a mistake bound of*

$$O\left(\min\left\{d(1-\gamma)^{-1} \cdot k_{in}k_{out}\ln(k_{in}k_{out}), |\mathcal{H}|\right\}\right).$$

We design two separate algorithms to achieve the bounds of $O(|\mathcal{H}|)$ (in Lemma 4.2) and $O\left((1-\gamma)^{-1} \cdot k_{\text{in}}k_{\text{out}}\ln(k_{\text{in}}k_{\text{out}})d\right)$ (by combining Lemma 4.3 and Corollary 3.2) respectively.

While achieving $O(|\mathcal{H}|)$ is usually trivial in most online learning problems (including our Revealed-Arb setting), it becomes subtle in the $\gamma$-Weighted setting. Typically, this linear guarantee can be obtained by trying each hypothesis in $\mathcal{H}$ and discarding it once it makes a mistake. However, in the $\gamma$-Weighted setting, even if we have identified the optimal classifier $h^\star$ and used it as the current classifier $h_t$, it may still make a mistake since the agent is not necessarily best responding to $h_t$.

To achieve this linear guarantee, we refine the approach of trying all hypotheses as shown in Algorithm 2. Specifically, we predict $x$ as positive if any existing hypothesis labels $x$ as positive. If we make a false positive mistake, we can eliminate one hypothesis. A false negative mistake can occur only if a false positive mistake happened in the previous round. Thus, we obtain the following guarantee. The proof of Lemma 4.2 is deferred to Appendix B.2.

---

**Algorithm 2** Conservative algorithm for $\gamma$-Weighted

---

1: **Initialization:** expert set $E = \mathcal{H}$
2: **for** $t = 1, 2, \ldots$ **do**
3:    Prediction: at each point $x$, $h_t(x) = 1$ if and only if there exists $h \in E$ such that $h(x) = 1$.
4:    Update: //*when observe a mistake at $v_t$*
5:    **if** $y_t = 0$ **then**
6:       $E \leftarrow E \setminus \{h : h(v_t) = 1\}$
7:    **end if**
8: **end for**

---

**Lemma 4.2.** *For any hypothesis class $\mathcal{H}$, in the strategic realizable setting, when the agents best-respond to the $\gamma$-weighted sum of the history $\widetilde{h}_t^\gamma$, Algorithm 2 achieves a mistake bound of $O(|\mathcal{H}|)$.*

To achieve the second mistake bound of $O\left(d(1-\gamma)^{-1} \cdot k_{\text{in}}k_{\text{out}}\ln(k_{\text{in}}k_{\text{out}})\right)$, we design a reduction to Revealed-Arb as shown in Algorithm 3. Given any algorithm $\mathcal{A}$ with a mistake bound guarantee in the Revealed-Arb setting, we update $\mathcal{A}$ and implement a new classifier only after a sufficiently large number of repeated mistakes, say $\Phi$ repeated mistakes. When the implemented classifier remains the same $h$ for at least $\Phi$ repeated mistakes, the weighted average of historical classifiers will be close to $h$. Consequently, when agents best respond to $\widetilde{h}_t^\gamma$, they are in fact best responding to $h_t$. However, as mentioned at the beginning of Section 3, tie-breaking in this setting is adversarial.

---

**Algorithm 3** Reduction from $\gamma$-Weighted to Revealed-Arb

---

1: **Input:** Parameters $\gamma$; an online learning algorithm $\mathcal{A}$ that achieves mistake bound $\mathcal{M}_{\text{Revealed-Arb}}(\mathcal{A})$ for Revealed-Arb agents.
2: **Initialization:** Update frequency: $\Phi \leftarrow \lceil \ln(\frac{1}{3})/\ln(\gamma) \rceil + 1$.
   Number of mistakes since the most recent update: $\phi \leftarrow 0$.
   Clock (current time step) of algorithm $\mathcal{A}$: $i \leftarrow 1$.
3: **for** $t = 1, 2, \ldots$ **do**
4:    Commitment: Commit to classifier $h_t \leftarrow h_i^{\mathcal{A}}$. //Use $\mathcal{A}$'s $i$-th output.
5:    Prediction: Agent $(x_t, y_t)$ arrives, manipulates to $v_t$, and receives label $h_t(v_t)$.
6:    Update: //*when we make a mistake at the observed node $v_t$*
7:    **if** $h_t(v_t) \neq y_t$ **then**
8:       $\phi \leftarrow \phi + 1$.
9:       **if** $\phi == \Phi$ **then**
10:          //*If the learner has made $\Phi$ mistakes since the last update of $\mathcal{A}$*
11:          Update $\mathcal{A}$: Feed $\mathcal{A}$ with observation $(v_t, y_t)$; $i \leftarrow i + 1$.
12:          $\phi \leftarrow 0$. //*Reset the mistake counter*
13:       **end if**
14:    **end if**
15: **end for**

---

By running Algorithm 3, the total number of mistakes is bounded by the mistake bound of $\mathcal{A}$ multiplied by the number of mistakes between each update of $\mathcal{A}$, which is $\Phi$.

**Lemma 4.3** (Mistake bound of Algorithm 3)**.** *Let $\mathcal{M}_{\text{Revealed-Arb}}(\mathcal{A}, \mathcal{H})$ be an upper bound on the number of mistakes that $\mathcal{A}$ makes on any strategic realizable sequence of agents in the Revealed-Arb setting. Then the $\gamma$-Weighted setting, i.e., $v_t \in \mathsf{BR}_{\widetilde{h}_t^\gamma}(x_t)$ for all $t \geq 1$, the mistake bound of Algorithm 3 satisfies*

$$\mathcal{M}_{\gamma\text{-Weighted}}(\mathcal{H}) \leq \left(\left\lceil \frac{\ln(\frac{1}{3})}{\ln(\gamma)} \right\rceil + 1\right) \cdot \mathcal{M}_{\text{Revealed-Arb}}(\mathcal{A}, \mathcal{H}).$$

We defer the proof of Lemma 4.3 to Appendix B.1. Combining Lemma 4.3 and Corollary 3.2, we achieve the mistake bound of $O\left((1-\gamma)^{-1} \cdot k_{\text{in}} k_{\text{out}} \ln(k_{\text{in}} k_{\text{out}}) d\right)$.

## 4.2. Lower Bound

In this subsection, we will provide the lower bound on $\mathcal{M}_{\gamma\text{-Weighted}}$, which shows that both the $k_{\text{in}}$ factor and the $(1-\gamma)^{-1}$ factor are unavoidable in certain regimes. We defer the proof to Appendix B.3.

**Theorem 4.4** (Lower bound on $\mathcal{M}_{\gamma\text{-Weighted}}$). *For any $d \in \mathbb{N}$ and discount factor $\gamma \in (0,1)$, there exists a hypothesis class $\mathcal{H}$ with Littlestone dimension $d$ and a manipulation graph $G$ with $k_{in} = k_{out} = 2$, such that in the strategic realizable setting, when agents best-respond to $\widetilde{h}_t^\gamma$, any deterministic learning algorithm must suffer a mistake bound of*

$$\mathcal{M}_{\gamma\text{-Weighted}} = \Omega\left(\min\left\{d(1-\gamma)^{-1}, |\mathcal{H}|\right\}\right).$$

*In the special case of $\gamma \to 0$, for any $k_{in}, k_{out} \in \mathbb{N}$ where $k_{out} \geq (1 + \Omega(1)) \cdot k_{in}$, there exists an instance with maximum in (resp. out)-degree $k_{in}$ (resp. $k_{out}$), such that any deterministic learning algorithm must have*

$$\mathcal{M}_{\gamma\text{-Weighted}} = \Omega(d \cdot k_{in} \cdot k_{out}).$$

In Theorem 4.4's lower bound, the $k_{\text{in}}$ factor arises due to arbitrary tie-breaking, which is consistent with the lower bound in the Revealed-Arb setting. This component of the lower bound is established using a modified version of the instance used in the proof of Theorem 3.3.

On the other hand, the $(1-\gamma)^{-1}$ factor reflects the cost of having historically implemented incorrect hypotheses. In fact, this lower bound still holds even when agents adopt standard tie-breaking. The construction is based on replicating a simple star graph (with one center and two leaves) by $|\mathcal{H}|$ times, as illustrated in Figure 2. Each hypothesis $h_i \in \mathcal{H}$ labels the right leaf of the $i$-th star as positive and the left leaves of all the other stars as positive, i.e., $h_i = \mathbb{1}_{x = x_{i,R} \text{ or } x \in \{x_{j,L} | j \neq i\}}$.

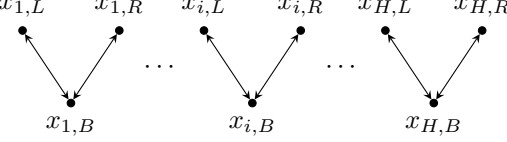

*Figure 2.* The graph used to establish the lower bound on $\mathcal{M}_{\gamma\text{-Weighted}}$ for general $\gamma \in (0,1)$.

Note that if the agent best responds to the current classifier (as in the Revealed-Std or Revealed-Arb setting), then this instance is easy to learn as $d, k_{\text{in}}, k_{\text{out}}$ are all constants. In fact, the learner can guarantee only a single mistake: by initially predicting the left leaf of each star as positive, the learner will eventually make a mistake on the star corresponding to the optimal hypothesis $h_{i^\star}$, at which point the learner can then switch to implementing $h_{i^\star}$ and makes no further mistakes.

However, the above approach incurs $(1-\gamma)^{-1}$ mistakes in the $\gamma$-Weighted setting. The adversary can repeatedly present correctly labeled agents to build up weight on $x_{i^\star, L}$ in the learner's $\gamma$-weighted classifier $\widetilde{h}_t^\gamma$, before inducing a mistake in the $i^\star$-th star. Once the learner identifies $i^\star$ through the mistake and attempts to switch to the correct prediction over the $i^\star$-th star by $\mathbb{1}_{x_{i^\star, R}}$, it must continue implementing this new hypothesis for $(1-\gamma)^{-1}$ rounds of repeated deployment of $\mathbb{1}_{x_{i^\star, R}}$ before the best response of agent $x_{i^\star, B}$ switches from $x_{i^\star, L}$ to $x_{i^\star, R}$. During this transition — which forces the agent to "forget" the influence of past classifiers — the learner continues to make mistakes.

In the full proof of Theorem 4.4 which we present in Appendix B.3, we show that this $(1-\gamma)^{-1}$ blow-up in mistakes is inevitable for any learning algorithm, unless $|\mathcal{H}|$ is small and the learner adopts a more conservative algorithm (e.g., Algorithm 2) to achieve mistake bound of $|\mathcal{H}|$.

## 5. Learning Agents

In recent years, repeated Stackelberg games with learning agents have become a rapidly growing area of study (see Brown et al. (2024); Haghtalab et al. (2024); Collina et al. (2024) for a non-exhaustive list). In such settings, in each round, the decision maker first selects a policy (corresponding to the classifier in strategic classification), and the agents subsequently choose an action (corresponding to which neighbor to manipulate to) by running an online learning algorithm instead of directly best responding. However, these results cannot be directly applied to strategic classification, as the agents' action sets, $N_{\text{out}}[x_t]$, vary over time. In contrast, the existing literature typically focuses on games such as auctions (Braverman et al., 2018; Cai et al., 2023) and contract design (Guruganesh et al., 2024), which assume a static action set for agents throughout the game.

Defining learning agents in strategic classification presents unique challenges, since we need meaningful performance guarantees for their learning algorithms. While regret (or variations such as swap-regret) is a standard performance measure in many game-theoretic settings, it does not translate directly to strategic classification for two reasons.

1. The agents' action sets, $N_{\text{out}}[x_t]$, vary over time, making the standard regret notions inapplicable because they usually assume a static action set. Although adopting *sleeping regret* (where an action/feature vector is only feasible in rounds where it is reachable from the agent's

initial feature vector) as a performance measure could address this issue, it remains an open direction due to the second challenge below.

2. Classical regret bounds often depend on the size of the action set. In graph-based strategic classification, actions correspond to nodes, and the total number of possible actions can be extremely large or even infinite, rendering traditional regret guarantees impractical or meaningless. Even in a geometric setting (e.g. Dong et al., 2018; Shen et al., 2024) where features are $d$-dimensional vectors in a Euclidean space and can be grouped based on geometric adjacency, Haghtalab et al. (2024) have highlighted a similar challenge of overcoming exponential dependency on the dimension $d$.

Though existing results cannot be directly applied to obtain performance guarantees for the agent's learning algorithm, we still aim to provide some insights into learning agents. A popular category of learning algorithms considered in this context is mean-based learning algorithms. Adapting the definition to the context of strategic classification, mean-based algorithms can be defined as follows.

**Definition 5.1** (Mean-Based Learning Algorithms (Braverman et al., 2018)). Let $\overline{h}_t = \frac{1}{t-1} \sum_{\tau=1}^{t-1} h_\tau$ be the empirical average of historical classifiers. An algorithm is $\eta$-mean-based if it is the case that whenever $\overline{h}_t(v') < \overline{h}_t(v) - \eta$ for $v, v' \in N_{\text{out}}[x_t]$, the probability that the algorithm chooses to manipulate to node $v'$ is at most $\eta$. An algorithm is called mean-based if it is $\eta$-mean-based for some $\eta = o(1)$.

Mean-based algorithms encompass most standard online learning algorithms, such as the Multiplicative Weights algorithm, the Follow-the-Perturbed-Leader algorithm, the $\varepsilon$-Greedy algorithm, and so on (Braverman et al., 2018). Our main $\gamma$-Weighted setting with $\gamma \to 1$, also falls in the mean-based class with $\eta = 0$. As shown in Section 4, the decision maker suffers $\Theta(|\mathcal{H}|)$ mistakes in this case. However, the situation can be worse for other mean-based algorithms. We observe that if $\eta$ is non-zero, the decision maker could suffer an infinite number of mistakes even in the strategic realizable setting.

More specifically, given the average of historical classifiers, $\overline{h}_t = \frac{1}{t-1} \sum_{\tau=1}^{t-1} h_\tau$, let $v_1 = \arg\max_{v \in N_{\text{out}}[x_t]} \overline{h}_t(v)$ and $v_2 = \arg\max_{v \in N_{\text{out}}[x_t] \setminus \{v_1\}} \overline{h}_t(v)$ be the empirically best and second best neighbors. For a mean-based algorithm $\mathcal{A}$, we define a monotonically decreasing function $\sigma_t^{\mathcal{A}} : [0,1] \mapsto [0,1]$ such that $\sigma_t^{\mathcal{A}}(\overline{h}_t(v_1) - \overline{h}_t(v_2))$ is a lower bound of the probability of choosing the second best neighbor $v_2$. For example, for Multiplicative Weights where the probability of choosing a neighbor $v$ is proportional to $\exp(\varepsilon_t \cdot \overline{h}_t(v))$ with learning rate $\varepsilon_t$, $\sigma_t^{\mathcal{A}}(z) \geq \frac{\exp(-\varepsilon_t \cdot (t-1) \cdot z)}{k_{\text{out}}}$. For $\varepsilon$-Greedy with exploration rate $\varepsilon_t$, $\sigma_t^{\mathcal{A}}(z) \geq \varepsilon_t / k_{\text{out}}$.

The function $\sigma_t^{\mathcal{A}}$ captures the bounded rationality of learning agents: as the gap between the best and second-best manipulations shrink, they have higher probability of choosing the suboptimal action even when the learner is already implementing the optimal classifier. This leads us to the following observation. See Appendix C for a formal version and the proof.

**Observation 5.2** (informal). *There exists a hypothesis class* $\mathcal{H} = \{h_1, h_2\}$ *of size* 2 *and a manipulation graph* $G$ *with* $k_{in}, k_{out} \leq 3$ *such that when agents run Multiplicative Weights/$\varepsilon$-Greedy with $\varepsilon_t$ set to be $\frac{1}{\sqrt{T}}$ or $\frac{1}{\sqrt{t}}$ (which are the most common choices of parameters), the decision maker will suffer infinite mistakes as $T$ goes to infinity.*

## 6. Discussion and Future Directions

In this paper, we investigate the question of whether decision-makers should make classifiers transparent in online strategic classification. We focus on a non-transparent setting in which agents do not observe the current classifier but manipulate their features based on a $\gamma$-weighted average of historical classifiers. We show that compared with the fully transparent setting, hiding the classifiers leads to an additional multiplicative factor of $(1 - \gamma)^{-1}$ or $k_{\text{in}}$ in the decision-maker's mistake bound. Notably, the $k_{\text{in}}$ factor is unavoidable even in the transparent setting when the agent switch from standard to adversarial tie-breaking among multiple best responses. We complement these lower bounds with algorithms whose mistake bounds grow by at most a $\widetilde{O}((1 - \gamma)^{-1} \cdot k_{\text{in}})$ factor relative to the the transparent setting.

There remains several open directions for future work. An immediate question is to close the gap between upper and lower bounds. While our lower bound shows the $k_{\text{in}}$ and $(1 - \gamma)^{-1}$ factors are individually unavoidable in different regimes, it remains to extent our construction in Theorem 4.4 to simultaneously deal with any combinations of $k_{\text{in}}$ and $\gamma$. Another open problem is extending our analysis to *randomized algorithms* (where the adversary's choice of the agent sequence cannot depend on the realized classifiers) or the *agnostic setting* (where the optimal classifier makes nonzero mistakes). The characterizations for both setting are particularly challenging and has remained open even in the standard transparent setting (Ahmadi et al., 2023; 2024; Cohen et al., 2024a). Finally, as discussed in Section 5, it remains open to consider other models of learning agents with nontrivial performance guarantee for both the decision-maker and the agents. We believe that *sleeping regret* is a natural candidate for agent's performance measure, but deriving meaningful sleeping regret bounds is challenging due to the large action space.

## Impact Statement

This work is purely theoretical. We do not foresee any immediate societal consequences.

## Acknowledgments

We thank Avrim Blum for the helpful discussions. HS was supported by Harvard CMSA.

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

# A. Omitted Proofs from Section 3

## A.1. Proof of Theorem 3.1

**Theorem 3.1** (Upper Bound on $\mathcal{M}_{\text{Revealed-Arb}}$). *Let $M^{ns}(\mathcal{A}, \mathcal{H})$ be the mistake bound of $\mathcal{A}$ in the standard (non-strategic) online learning setting when the inputs are (non-strategic) realizable by $\mathcal{H}$. In online strategic classification where the agent sequence is strategic realizable under $\mathcal{H}$, and each agent best responds to $h_t$ with adversarial tie-breaking, Algorithm 1 achieves*

$$\mathcal{M}_{\text{Revealed-Arb}}(\mathcal{H}) = O(k_{in} \cdot k_{out} \cdot \ln(k_{in}k_{out})) \cdot M^{ns}(\mathcal{A}, \mathcal{H}).$$

*Proof of Theorem 3.1.* Suppose $\mathcal{A}$ is the standard online learning algorithm that is input to Algorithm 1, and let $M$ be the number of mistakes that $A$ makes on any realizable sequence. To keep track of the algorithm's progress, we use $W_t$ to denote the total weight of experts at the beginning of round $t$, i.e.,

$$W_t := \sum_{A \in E} w_A^{(t)}.$$

We first show that whenever Algorithm 1 makes a mistake at round $t$, the total weight satisfies

$$W_{t+1} \leq \left(1 - \frac{1}{4(k_{\text{out}} + 1)(k_{\text{in}} + 1)}\right) \cdot W_t.$$

We analyze the following two cases.

**Case 1: False positive mistake.** In this case, the agent is labeled as $h_t(v_t) = 1$ after manipulation, but the true label is $y_t = 0$. Since the true label is 0, the realizability of input sequence $S$ implies that the optimal hypothesis $h^\star$ should label the entire neighborhood $N_{\text{out}}[x_t]$—including $v_t$—as 0. Therefore, we proceed by updating all experts that predict $A(v_t) = 1$ by feeding them with the example $(v_t, 0)$, and halving their weights. Since $h_t(v_t) = 1$, according to the prediction rule in Line 4, we must have

$$\sum_{A \in E : A(v_t) = 1} w_A^{(t)} \geq \frac{W_t}{2(k_{\text{out}} + 1)(k_{\text{in}} + 1)}.$$

Therefore, the updated total weight satisfies

$$\begin{aligned}
W_{t+1} &= W_t - \frac{1}{2} \sum_{A \in E : A(v_t) = 1} w_A^{(t)} \\
&\leq \left(1 - \frac{1}{4(k_{\text{out}} + 1)(k_{\text{in}} + 1)}\right) \cdot W_t
\end{aligned}$$

**Case 2: False negative mistake.** In this case, the agent is predicted as $h_t(v_t) = 0$, but the true label is $y_t = 1$. This implies that our learner $h_t$ labeled the entire out neighborhood $N_{\text{out}}[x_t]$ with 0, otherwise, if there is an $x' \in N_{\text{out}}[x_t]$ with $h_t(x') = 1$, $x_t$ would have manipulated to $x'$ and received a positive label. In addition, since the sequence of agents is realizable, we know that $h^\star(\text{BR}_{h^\star}(x_t)) = 1$, i.e., $h^\star$ must label some node in $N_{\text{out}}[x_t]$ as 1.

However, the main challenge in this adversarial tie-breaking setting is that we do not observe the true label $x_t$. Instead, we know that $x_t$ can be any of the in-neighbors of $v_t$ that satisfies $h_t(N_{\text{out}}[x]) = 0$ (which means $h_t$ labels all nodes in $N_{\text{out}}[x]$ as 0). Thus, we can construct a set of candidates for $x_t$ as follows (as described in Line 9 of Algorithm 1):

$$\widetilde{\mathcal{X}}_t := \{x \in N_{\text{in}}[v_t] \mid h_t(N_{\text{out}}[x]) = 0\}$$

Since $\widetilde{\mathcal{X}}_t$ contains all possible choices of $x_t$, any expert that labels the union of their out-neighborhood (i.e., $\widetilde{\mathcal{V}}_t := \cup_{x \in \widetilde{\mathcal{X}}_t} N_{\text{out}}[x_t]$) as all negative must be wrong. Based on our prediction rule, we can show that the total weight of such

experts should take up a significant fraction of $W_t$:

$$\sum_{A \in E: \forall x' \in \widetilde{\mathcal{V}}_t, A(x')=0} w_A^{(t)}$$

$$\geq W_t - \sum_{x' \in \widetilde{\mathcal{V}}_t} \sum_{A \in E: A(x')=1} w_A^{(t)}$$

$$\overset{(a)}{\geq} \left(1 - \frac{|\widetilde{\mathcal{V}}_t|}{2(k_{\text{out}}+1)(k_{\text{in}}+1)}\right) W_t$$

$$\overset{(b)}{\geq} \left(1 - \frac{k_{\text{in}} \cdot k_{\text{out}}}{2(k_{\text{out}}+1)(k_{\text{in}}+1)}\right) W_t$$

$$\geq \frac{1}{2} W_t.$$

In the above inequalities, step (a) is because all nodes in $\widetilde{\mathcal{V}}_t$ are labeled as 0 by $h_t$, which, according to the prediction rule in Line 4, implies that for any $x' \in \widetilde{\mathcal{V}}_t$, we have $\sum_{A \in E: A(x')=1} w_A^{(t)} < \frac{W_t}{2(k_{\text{out}}+1)(k_{\text{in}}+1)}$. Step (b) is because the size of $\widetilde{\mathcal{V}}_t$ is at most $k_{\text{in}} \cdot k_{\text{out}}$.

According to the algorithm, we split each of such $A$ into $|\widetilde{\mathcal{V}}_t|$ experts—i.e., $\{A(x', 1) \mid x' \in \widetilde{\mathcal{V}}_t\}$—and split the weight $\frac{1}{2} w_A$ equally among them. Thus, we have $W_{t+1} \leq (1 - \frac{1}{4})W_t \leq \left(1 - \frac{1}{4(k_{\text{out}}+1)(k_{\text{in}}+1)}\right) \cdot W_t$.

**Deriving the final mistake bound.** We have proved that whenever Algorithm 1 makes a mistake, the total weight $W_t$ is reduced by a multiplicative factor of $\frac{1}{4(k_{\text{out}}+1)(k_{\text{in}}+1)}$. Let $N$ denote the total number of mistakes that Algorithm 1 makes. Then we have total weight in the final round is at most $\left(1 - \frac{1}{4(k_{\text{out}}+1)(k_{\text{in}}+1)}\right)^N$.

On the other hand, note that there exists an expert $A^\star \in E$ that is an execution of $\mathcal{A}$ with trajectory $(v_t', y_t)$ during the mistake rounds, where $v_t' \in \text{BR}_{h^\star}(x_t)$. Since the agent sequence $(x_t, y_t)_{t \geq 1}$ is strategic realizable, we have $h^\star(\text{BR}_{h^\star}(x_t)) = y_t$, which implies $h^\star(v_t') = y_t$. In other words, the inputs of $A^\star$ are realizable under $h^\star$. According to the mistake bound assumption of $\mathcal{A}$, $A^\star$ makes at most $M$ mistakes.

Now we analyze the weight of $A^\star$. At each mistake round $t$, if the $A^\star$ makes a false positive mistake, its weight is reduced by half. If $A^\star$ made a false negative mistake, it is split into a few experts, one of which is fed by $(v_t', y_t)$ where $v_t' \in \text{BR}_{h^\star}(x_t)$. This specific new expert's weight is reduced by at most $\frac{1}{2(k_{\text{out}}+1)(k_{\text{in}}+1)}$. Thus, since $A^\star$ makes at most $M$ mistakes, its weight at the final round will be at least $\left(\frac{1}{2(k_{\text{out}}+1)(k_{\text{in}}+1)}\right)^M$.

Combining these two observations, we have

$$\left(1 - \frac{1}{4(k_{\text{out}}+1)(k_{\text{in}}+1)}\right)^N \geq \left(\frac{1}{2(k_{\text{out}}+1)(k_{\text{in}}+1)}\right)^M,$$

which yields the mistake bound of Algorithm 1:

$$N \leq 4(k_{\text{out}}+1)(k_{\text{in}}+1)\ln(2(k_{\text{out}}+1)(k_{\text{in}}+1))M.$$

The proof is thus complete. □

### A.2. Proof of Theorem 3.3

**Theorem 3.3** (Lower bound on $\mathcal{M}_{\text{Revealed-Arb}}$)**.** *For any $d, k_{in}, k_{out} \in \mathbb{N}$ with $k_{out} \geq k_{in}$, there exists a hypothesis class $\mathcal{H}$ with Littlestone dimension $d$, and manipulation graph $G$ with maximum in (resp. out)-degree $k_{in}$ (resp. $k_{out}$), such that in the strategic realizable setting, any deterministic learning algorithms must suffer*

$$\mathcal{M}_{\text{Revealed-Arb}}(\mathcal{H}) = \Omega(k_{in} \cdot k_{out} \cdot d).$$

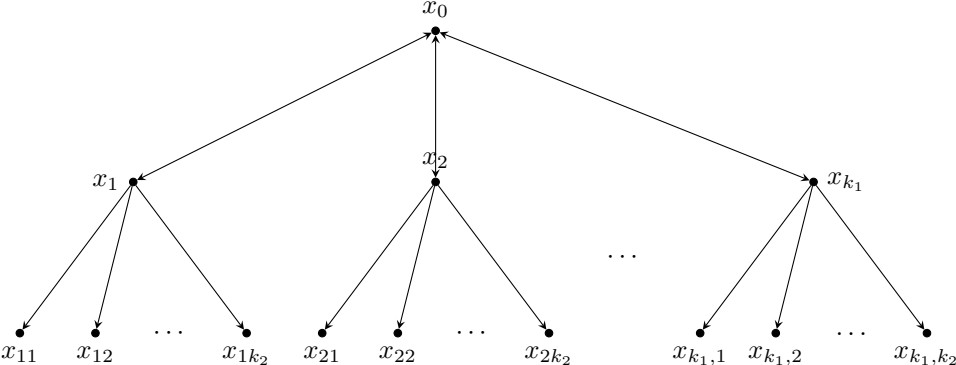

*Figure 3.* Example for lower bound construction for the Revealed-Arb setting.

*Proof of Theorem 3.3.* Consider a graph with nodes $x_0, x_1, \ldots, x_{k_1}, x_{11}, \ldots, x_{1k_2}, \ldots, x_{k_1,1}, \ldots, x_{k_1,k_2}$ as shown in Fig 3 with $k_2 > k_1$. The maximum out-degree is $k_2$ and the maximum in-degree is $k_1$. The hypothesis class contains all singletons over the leaves. Suppose the target hypothesis is $h_{i^\star, j^\star}$. Then only agents with original feature vectors at $x_{i^\star}$ and $x_{i^\star, j^\star}$ are labeled as positive. For any deterministic algorithm, at time $t$, there are the following cases:

- If $h_t$ is all-negative, the adversary can pick $x_t = x_{i^\star}$ and tie-breaking favors $x_0$. Then we only observe $v_t = x_0, y_t = 1$ and $\widehat{y}_t = 0$. $h_t$ makes a mistake but learn nothing.

- If $h_t$ labels $x_0$ as positive, the adversary can pick $x_t = x_0$ and tie-breaking favors $x_0$. Then we observe $v_t = x_0$, $y_t = 0$ and $\widehat{y}_t = 1$. Again, $h_t$ makes a mistake but learn nothing.

- If $h_t$ labels any node $x_i$ in the second layer as positive, the adversary can pick $x_t = x_0$ and tie-breaking favors $x_i$. Then we observe $v_t = x_i, y_t = 0$ and $\widehat{y}_t = 1$. Again, $h_t$ makes a mistake but we learn nothing.

- Hence, $h_t$ can only label some leaves as positive. When it labels any leaf $x_{ij}$ other than the target one as positive, the adversary picks $x_t = x_{ij}$ being that leaf and tie-breaking favors $x_{ij}$. $h_t$ makes a mistake and learns that $x_{ij}$ is not the target leaf.

Therefore, the algorithm will make $k_1 k_2$ mistakes. The Littlestone dimension of this hypothesis class is 1. We can extend the result to Littlestone dimension being $d$ by making $d$ independent copies of this example. □

## B. Omitted Proofs from Section 4

### B.1. Proof of Lemma 4.3

**Lemma 4.3** (Mistake bound of Algorithm 3). *Let $\mathcal{M}_{\text{Revealed-Arb}}(\mathcal{A}, \mathcal{H})$ be an upper bound on the number of mistakes that $\mathcal{A}$ makes on any strategic realizable sequence of agents in the* Revealed-Arb *setting. Then the $\gamma$-Weighted setting, i.e., $v_t \in \mathsf{BR}_{\overline{h}_t^\gamma}(x_t)$ for all $t \geq 1$, the mistake bound of Algorithm 3 satisfies*

$$\mathcal{M}_{\gamma\text{-Weighted}}(\mathcal{H}) \leq \left( \left\lceil \frac{\ln(\frac{1}{3})}{\ln(\gamma)} \right\rceil + 1 \right) \cdot \mathcal{M}_{\text{Revealed-Arb}}(\mathcal{A}, \mathcal{H}).$$

*Proof of Lemma 4.3.* Let $\mathcal{A}'$ be the algorithm obtained by applying the reduction in Algorithm 3 to $\mathcal{A}$. Let $(t_i)_{i \geq 1}$ be the time steps in which line 11 of Algorithm 3 is called. The key to establishing this lemma is to show that, for any $t = t_i$, the agent's response will best respond to the current $h_t$ with arbitrary tie-breaking by manipulating to $v_t = \mathsf{BR}_{\widetilde{h}_t^\gamma}(x_t)$.

To see this, note that the condition in Line 9 ensures that $\mathcal{A}'$ has made at least $\Phi = \lceil \ln(\frac{1}{3})/\ln(\gamma) \rceil + 1$ mistakes in the time interval $[t_{i-1} + 1, t_i]$, which implies that $t_i - (t_{i-1} + 1) \geq \Phi - 1$. Moreover, during this time interval, $\mathcal{A}'$ has been using

the same classifier $h_{i-1}^{\mathcal{A}}$. Therefore, at time $t = t_i$, the $\gamma$-discounted average classifier can be written as

$$
\begin{aligned}
\widetilde{h}_t^\gamma &= \frac{1-\gamma}{1-\gamma^{t-1}} \cdot \sum_{\tau=1}^{t-1} \gamma^{\tau-1} h_{t-\tau} \\
&= \frac{1-\gamma}{1-\gamma^{t-1}} \cdot \sum_{\tau=1}^{\Phi-1} \gamma^{\tau-1} h_t + \frac{1-\gamma}{1-\gamma^{t-1}} \cdot \sum_{\tau=\Phi}^{t-1} \gamma^{\tau-1} h_{t-\tau} &&(h_{t-\tau} = h_t \text{ for } \tau \le \Phi - 1) \\
&= \frac{1-\gamma^{\Phi-1}}{1-\gamma^{t-1}} \cdot h_t + \frac{\gamma^{\Phi-1}-\gamma^{t-1}}{1-\gamma^{t-1}} \cdot \widetilde{h} &&\left(\widetilde{h} := \tfrac{1-\gamma}{\gamma^{\Phi-1}-\gamma^{t-1}} \sum_{\tau=\Phi}^{t-1} \gamma^{\tau-1} h_{t-\tau}\right) \\
&= (1-\varepsilon_t) \cdot h_t + \varepsilon_t \cdot \widetilde{h}. &&\left(\varepsilon_t := \tfrac{\gamma^{\Phi-1}-\gamma^{t-1}}{1-\gamma^{t-1}}\right)
\end{aligned}
$$

In the above equation, we have

$$
\varepsilon_t = \frac{\gamma^{\Phi-1}-\gamma^{t-1}}{1-\gamma^{t-1}} \le \gamma^{\Phi-1} \le \gamma^{\ln(\frac{1}{3})/\ln(\gamma)} = \frac{1}{3}.
$$

This implies that $\widetilde{h}_t^\gamma$ is already very close to the true $h_t$. If $h_t(\mathsf{BR}_{h_t}(x_t)) = 1$, then we claim $\mathsf{BR}_{\widetilde{h}_t^\gamma}(x_t) \subset \mathsf{BR}_{h_t}(x_t)$ because for any $x_t$'s neighbor $v \notin \mathsf{BR}_{h_t}(x_t)$, it holds that

$$
\begin{aligned}
\widetilde{h}_t^\gamma(v) &= (1-\varepsilon_t) \cdot h_t(v) + \varepsilon_t \cdot \widetilde{h}(v) = \varepsilon_t \cdot \widetilde{h}(v) \\
&\le \varepsilon_t < 1 - \varepsilon_t \le \max_{v' \in N_{\text{out}}[x_t]} \widetilde{h}_t^\gamma(v').
\end{aligned}
$$

If $h_t(\mathsf{BR}_{h_t}(x_t)) = 0$, then $\mathsf{BR}_{\widetilde{h}_t^\gamma}(x_t)$ being some $v \in N_{\text{out}}[x_t]$ can be viewed as breaking tie adversarially for best responding to $h_t$.

Per guarantee of algorithm $\mathcal{A}$, we conclude that the number of mistakes that $\mathcal{A}'$ makes on the subsequence $S' = (x_{t_i}, y_{t_i})_{i \ge 1}$ is at most $\mathcal{M}_{\text{Revealed-Arb}}(\mathcal{A}, \mathcal{H})$.

Finally, it remains to bound the mistakes on the entire sequence $S$. Since $\mathcal{A}$ calls $\mathcal{A}'$ once per $\Phi$ mistakes, the total number of mistakes is at most

$$
\begin{aligned}
\mathcal{M}_{\gamma\text{-Weighted}}(\mathcal{A}', \mathcal{H}) &\le \Phi \cdot \mathcal{M}_{\text{Revealed-Arb}}(\mathcal{A}, \mathcal{H}) \\
&= (\lceil \ln(1/3)/\ln(\gamma) \rceil + 1) \cdot \mathcal{M}_{\text{Revealed-Arb}}(\mathcal{A}, \mathcal{H}).
\end{aligned}
$$

The proof is thus complete. $\qquad\square$

### B.2. Proof of Lemma 4.2

**Lemma 4.2.** *For any hypothesis class $\mathcal{H}$, in the strategic realizable setting, when the agents best-respond to the $\gamma$-weighted sum of the history $\widetilde{h}_t^\gamma$, Algorithm 2 achieves a mistake bound of $O(|\mathcal{H}|)$.*

*Proof of Lemma 4.2.* First we show that $h^\star$ is always in $E$. For any $y_t = 0$, $h^\star(v_t) = 0$ because $\max_{v \in N_{\text{out}}[x_t]} h^\star(v) = 0$. Then $h^\star$ will never be removed from $E$.

Next we show the number of false negative error is no larger than the number of false positive error. Suppose we make a false negative mistake at time $t$, i.e., $y_t = 1$ and $h_t(v_t) = 0$. For $v \in \mathsf{BR}_{h^\star}(x_t)$, $h_1(v) = \cdots = h_{t-1}(v) = 1$ because $h^\star(v) = 1$ and $h^\star \in E$ at time $1, \cdots, t-1$. Then we have $\widetilde{h}_t^\gamma(v) = 1$ by the definition of $\widetilde{h}_t^\gamma$. Moreover it holds that $\widetilde{h}_t^\gamma(v_t) \ge \widetilde{h}_t^\gamma(\mathsf{BR}_{h^\star}(v)) = 1$. So $h_1(v_t), \cdots, h_{t-1}(v_t)$ must all be positive. $h_t(v_t) = 0$ means that $E$ is updated at time $t-1$ otherwise $h_{t-1}$ will be the same as $h_t$. So a false negative mistake can only occur right after a false positive mistake.

Every time we make a false positive mistake, at least one hypothesis will be removed from $E$ otherwise $v_t$ won't be classified as positive. So the number of false positive mistakes is upper bounded by the number of candidate hypothesis $|\mathcal{H}|$ and the number of total mistakes is upper bounded by $2|\mathcal{H}|$.

$\qquad\square$

## B.3. Proof of Theorem 4.4

**Theorem 4.4** (Lower bound on $\mathcal{M}_{\gamma\text{-Weighted}}$)**.** *For any $d \in \mathbb{N}$ and discount factor $\gamma \in (0,1)$, there exists a hypothesis class $\mathcal{H}$ with Littlestone dimension $d$ and a manipulation graph $G$ with $k_{in} = k_{out} = 2$, such that in the strategic realizable setting, when agents best-respond to $\widetilde{h}_t^\gamma$, any deterministic learning algorithm must suffer a mistake bound of*

$$\mathcal{M}_{\gamma\text{-Weighted}} = \Omega\left(\min\left\{d(1-\gamma)^{-1}, |\mathcal{H}|\right\}\right).$$

*In the special case of $\gamma \to 0$, for any $k_{in}, k_{out} \in \mathbb{N}$ where $k_{out} \geq (1 + \Omega(1)) \cdot k_{in}$, there exists an instance with maximum in (resp. out)-degree $k_{in}$ (resp. $k_{out}$), such that any deterministic learning algorithm must have*

$$\mathcal{M}_{\gamma\text{-Weighted}} = \Omega(d \cdot k_{in} \cdot k_{out}).$$

*Proof of Theorem 4.4.* We prove this theorem by establishing the two lower bounds separately. First, adapting the construction in Theorem 3.3, we show that as $\gamma \to 0$, the freedom of arbitrary tie-breaking forces the decision-maker to suffer $d \cdot k_{\text{in}} \cdot k_{\text{out}}$ mistakes. Next, we use a different instance that exploits the agent's effective memory length to show that $\mathcal{M}_{\text{Revealed-Arb}} = \Omega\left(\min\{d(1-\gamma)^{-1}, |\mathcal{H}|\}\right)$. Combining these bounds completes the proof.

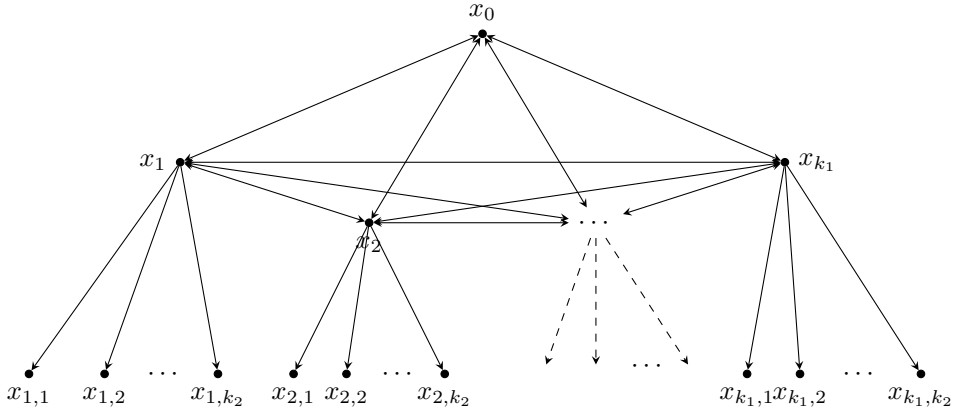

*Figure 4.* Lower bound construction when $\gamma \to 0$

**Part 1: $\mathcal{M}_{\gamma\text{-Weighted}} = \Omega(d \cdot k_{\textbf{in}} \cdot k_{\textbf{out}})$ as $\gamma \to 0$.** For this lower bound, consider the graph that is the same as that in Figure 3, but all nodes in the first layer (namely $x_1, x_2, \ldots, x_{k_1}$) are connected by a clique, as shown in Figure 4. This modification gives $k_{\text{out}} = k_1 + k_2$ and $k_{\text{in}} = k_2$. As in Theorem 3.3, we let the hypothesis class consist of all singletons over the leaves, i.e., $\mathcal{H} = \{h_{i,j} = \mathbb{1}_{x_{i,j}} \mid i \in [k_1], j \in [k_2]\}$. Suppose the target hypothesis is some unknown $h_{i^\star,j^\star} \in \mathcal{H}$. Under the strategic realizability assumption, only agents originally located at $x_{i^\star}$ or $x_{i^\star,j^\star}$ have positive true labels.

For $\gamma \to 0$, best responding to $\widetilde{h}_t^\gamma$ is equivalent to best responding to $h_{t-1}$. We will construct an adversary that induces at least one mistake in every two rounds in the first $2k_1 k_2$ rounds. This will establish a lower bound of $\Omega(k_{\text{in}} \cdot (k_{\text{out}} - k_{\text{in}})) = \Omega(k_{\text{in}} \cdot k_{\text{out}})$ under the assumption that $k_{\text{out}} \geq (1 + \Omega(1)) \cdot k_{\text{in}}$.

Consider two consecutive rounds $t - 1$ and $t$ (where $t = 2, 4, \ldots, 2k_1 k_2$). If the learner already makes a mistake at round $t - 1$, then a mistake have been induced across these two rounds. Otherwise, we show below how to force a mistake in round $t$ by performing a case discussion of both $h_{t-1}$ — which the agent $x_t$ best responds to — and the current classifier $h_t$. We let the initial version space contain all hypotheses.

1. If $h_{t-1}$ labels $x_0$ by positive, then consider two sub-cases. If $h_t(x_0) = 1$, the adversary can pick $(x_t, y_t) = (x_0, 0)$, and make the tie-breaking favor $x_0$. This forces $h_t$ to make a false positive mistake but learns nothing.

   On the other hand, if $h_t(x_0) = 0$, then the adversary picks some $(i^\star, j^\star)$[2] consistent with the current version space,

---

[2]Since the learner's algorithm is deterministic, the adversary can simulate the interaction between the algorithm and the adversary, and pick the hypothesis $h_{i^\star,j^\star}$ that is the last one to be removed from the version space. This makes the realizable hypothesis consistent across all rounds.

sets $(x_t, y_t) = (x_{i^\star}, 1)$, and again makes the tie-breaking favor $x_0$. As a result, the $t$-th agent will manipulate to $x_0$ and induces a false negative mistake while revealing no information.

2. If $h_t$ labels any leaf node $x_{i,j}$ as positive, then the adversary picks $(x_t, y_t) = (x_{i,j}, 0)$, and removes hypothesis $h_{i,j}$ from the version space if it has not been removed yet. This induces a false positive mistake. Since the learner made no mistake in round $t - 1$, we can assume that $h_{t-1}$ labels all leaf nodes as negative.

3. We are left with the case where $h_{t-1}(x_0) = -1$ and $h_t$ labels all leaf nodes as negative. In addition, from case 2, it is without loss of generality to assume that $h_{t-1}(x_{i,j}) = -1$ for all $i \in [k_1]$ and $j \in [k_2]$, as otherwise the learner could have induced a false positive mistake in round $t - 1$ already. Therefore, only the middle layer $\{x_1, \ldots, x_{k_1}\}$ can be potentially labeled as positive by $h_{t-1}$. Due to arbitrary tie-breaking and the added clique, we can assume that all nodes in $\{x_0, x_1, \ldots, x_{k_1}\}$ manipulates to the same node $x_i$ where $i \in \arg\max_{i \in [k_1]} h_{t-1}(x_i)$ in response to $h_{t-1}$. Now consider the following two subcases:

   If $h_t(x_i) = 1$, then the adversary can choose $(x_t, y_t) = (x_0, 0)$, which will manipulate to $x_i$ and induce a false positive mistake. If $h_t(x_i) = 0$, then the adversary chooses $(x_t, y_t) = (x_{i^\star}, 1)$ from the version space[3] and induces a false negative mistake, while revealing no information about $i^\star$.

   Note that the proof of this case crucially relies on the additional clique structure. Without this clique, the learner can set $h_{t-1}$ to predict all middle-layer nodes $(x_1, \ldots, x_{k_1})$ as positive, and set $h_t$ to predict all nodes as negative. This would cause all leaf nodes $x_{i,j}$ to manipulate to their parent $x_i$, while the middle-layer nodes stay put. In this case, for the adversary to induce a mistake, it would have to reveal information about $i^\star$, which would allow the learner to either eliminate up to $k_2$ hypotheses in a single round.

Since the version space initially contains $k_1 \cdot k_2$ classifiers, the above construction can be used to induce at least $k_1 \cdot k_2 - 1$ mistakes before the version space becomes empty. This establishes the $\Omega(k_1 \cdot k_2)$ mistake bound when $d = 1$. For general $d$, we can create $d$ independent copies of the same instance to obtain a bound of $\Omega(k_1 \cdot k_2 \cdot d)$.

**Part 2: For any** $\gamma \in (0, 1)$, $\mathcal{M}_{\gamma\text{-Weighted}} = \Omega\left(\min\{d(1 - \gamma)^{-1}, |\mathcal{H}|\}\right)$. For simplicity, we consider the agents best respond to unnormalized weighted sum of history

$$\overline{h}_t^\gamma = \sum_{\tau=0}^{t-2} \gamma^\tau h_{t-1-\tau}$$

in this proof because the agents will make the same manipulation for $\widetilde{h}_t^\gamma$ and $\overline{h}_t^\gamma$. We first provide an example for hypothesis class with Littlestone dimension being 1. Consider the graph with $x_{1,B}, x_{1,L}, x_{1,R}, \ldots, x_{H,B}, x_{H,L}, x_{H,R}$ as shown in Fig 2. There are $H$ classifier in the hypothesis class $\mathcal{H}$ with $h_i$ defined as $h_i(x_{j,B}) = 0$ for all $j$; $h_i(x_{j,L}) = 1, h_i(x_{j,R}) = 0$ for any $j \neq i$; and $h_i(x_{i,L}) = 0, h_i(x_{i,R}) = 1$. Suppose the target hypothesis is $h_{i^\star}$.

At each round $t$, there are three cases regarding the weighted sum of historical classifiers $\overline{h}_t^\gamma$:

1. If there exists $i \neq i^\star$ such that $\overline{h}_t^\gamma(x_{i,B}) = \max\{\overline{h}_t^\gamma(x_{i,B}), \overline{h}_t^\gamma(x_{i,L}), \overline{h}_t^\gamma(x_{i,R})\}$, we consider the following two cases of $h_t$.

   (a) If $h_t$ labels $x_{i,B}$ as negative, the adversary can pick $x_t = x_{i,B}$. The agent will always stay at $x_{i,B}$ under standard tie-breaking rule. We will observe $v_t = x_{i,B}, y_t = 1$ and $\widehat{y}_t = 0$. $h_t$ makes a mistake but learns nothing.

   (b) If $h_t$ labels $x_{i,B}$ as positive and $\overline{h}_t^\gamma(x_{i,B}) > \overline{h}_t^\gamma(x_{i,R})$, the adversary can pick $x_t = x_{i,R}$. The agent will manipulate to $x_{i,B}$. $h_t$ makes a mistake but learns nothing.

   (c) If $\overline{h}_t^\gamma(x_{i,B}) = \overline{h}_t^\gamma(x_{i,R}) \geq \overline{h}_t^\gamma(x_{i,L})$ and $h_t$ labels both $x_{i,B}$ and $x_{i,R}$ as positive, the adversary can pick $x_t = x_{i,R}$. The agent will stay at $x_{i,R}$. $h_t$ makes a mistake and we can eliminate $h_i$.

   (d) If $\overline{h}_t^\gamma(x_{i,B}) = \overline{h}_t^\gamma(x_{i,R}) \geq \overline{h}_t^\gamma(x_{i,L})$ and $h_t$ labels $x_{i,B}$ as positive and $x_{i,R}$ as negative, we can't force a mistake at this round. However, $\overline{h}_{t+1}^\gamma(x_{i,B})$ will be strictly larger than $\overline{h}_{t+1}^\gamma(x_{i,R})$ because $h_t(x_{i,B}) > h_t(x_{i,R})$. Then we force a mistake in round $t + 1$ without revealing any information like we did in (a) and (b) above.

---

[3]Again, assume that $h_{i^\star, j^\star}$ is the last hypothesis that remains in the version space.

2. If there exists $i \neq i^\star$ such that $\overline{h}_t^\gamma(x_{i,R}) = \max\{\overline{h}_t^\gamma(x_{i,B}), \overline{h}_t^\gamma(x_{i,L}), \overline{h}_t^\gamma(x_{i,R})\}$ but $\overline{h}_t^\gamma(x_{i,B}) < \overline{h}_t^\gamma(x_{i,R})$, we consider the following two cases of $h_t$.

   (a) If $h_t$ labels $x_{i,R}$ as positive, the adversary can pick $x_t = x_{i,R}$. The agent will stay at $x_{i,R}$. We will observe $y_t = 0$ and $\widehat{y}_t = 1$. $h_t$ makes a mistake and we can eliminate $h_i$.

   (b) If $h_t$ labels $x_{i,R}$ as negative, the adversary can pick $x_t = x_{i,B}$. With probability over $1/2$, the agent will manipulate to $x_{i,R}$. We will observe $y_t = 1$ and $\widehat{y}_t = 0$. $h_t$ makes a mistake but we can't make any progress.

3. If $\overline{h}_t^\gamma(x_{i,L}) > \overline{h}_t^\gamma(x_{i,B})$ and $\overline{h}_t^\gamma(x_{i,L}) > \overline{h}_t^\gamma(x_{i,R})$ for all $i \neq i^*$, we consider the following cases of $h_t$.

   (a) If $h_t$ labels $x_{i,L}$ as negative, the adversary can pick $x_t = x_{i,B}$. The agent will manipulate to $x_{i,L}$. $h_t$ makes a mistake but learns nothing.

   (b) If $h_t$ labels $x_{i,L}, x_{i,B}, x_{i,R}$ all as positive, the adversary can pick $x_t = x_{i,R}$. If the agent stays at $x_{i,R}$, we make a mistake and can eliminate $h_i$. If the agent manipulates to $x_{i,B}$, we make a mistake but can't learn anything.

   (c) If $h_t$ labels $x_{i,L}$ and $x_{i,R}$ as positive and labels $x_{i,B}$ as negative, the adversary can pick $x_t = x_{i,B}$. The agent will manipulate to $x_{i,L}$. We don't make a mistake but we also learn nothing. The only benefit in this round is that $\gamma$-weighted-sum will receive more weight for $x_{i,R}$ than $x_{i,B}$ from $h_t$.

   (d) If $h_t$ labels $x_{i,L}$ as positive and $x_{i,R}$ as negative, the adversary will pick $x_t = x_{i,B}$. Again we don't make mistakes and learn nothing. But $\gamma$-weighted-sum will receive more weight for $x_{i,L}$ than $x_{i,R}$ from $h_t$.

Start from some time $T$. We hope to achieve one of the following goals at time $T' > T$ where $T'$ is some number we will choose later:

- There exists a feasible $i$ such that $\overline{h}_{T'}(\gamma)(x_{i,L}) > \overline{h}_{T'}(\gamma)(x_{i,R}) + \frac{1}{3(1-\gamma)}$.

- There exists a feasible $i$ such that $\overline{h}_{T'}(\gamma)(x_{i,R}) > \overline{h}_{T'}(\gamma)(x_{i,B})$.

- We make at least one mistake and the hypotheses we can eliminate is no more than the mistakes we make.

The third case will happen once one of 1(a-d), 2(a), 2(b), 3(a), 3(b) happens. So we would assume only 3(c) and 3(d) occur between time $T$ and $T'$.

For any feasible $i$ at time $T$, we define $A_i = \{T \leq t < T' | h_t(x_{i,L}) = 1, h_t(x_{i,R}) = 1, h_t(x_{i,B}) = 0\}$, $B_i = \{T \leq t < T' | h_t(x_{i,L}) = 1, h_t(x_{i,R}) = 0, h_t(x_{i,B}) = 1\}$ and $C_i = \{T \leq t < T' | h_t(x_{i,L}) = 1, h_t(x_{i,R}) = 0, h_t(x_{i,B}) = 0\}$. We further define $\Delta_A = \sum_{t \in A_i} \gamma^{T'-t}$, $\Delta_B = \sum_{t \in B_i} \gamma^{T'-t}$ and $\Delta_C = \sum_{t \in C_i} \gamma^{T'-t}$. Then we can have that

$$\overline{h}_{T'}(\gamma)(x_{i,L}) = \gamma^{T'-T}\overline{h}_T(\gamma)(x_{i,L}) + \Delta_A + \Delta_B + \Delta_C$$
$$= \gamma^{T'-T}\overline{h}_T(\gamma)(x_{i,L}) + \frac{1 - \gamma^{T'-T}}{1 - \gamma},$$
$$\overline{h}_{T'}(\gamma)(x_{i,R}) = \gamma^{T'-T}\overline{h}_T(\gamma)(x_{i,R}) + \Delta_A,$$
$$\overline{h}_{T'}(\gamma)(x_{i,B}) = \gamma^{T'-T}\overline{h}_T(\gamma)(x_{i,B}) + \Delta_B,$$

If neither of the two goals is achieved for $i$, then it must hold that

$$\frac{1}{3(1-\gamma)} \geq \overline{h}_{T'}(\gamma)(x_{i,L}) - \overline{h}_{T'}(\gamma)(x_{i,R})$$
$$= \gamma^{T'-T}\left(\overline{h}_T(\gamma)(x_{i,L}) - \overline{h}_T(\gamma)(x_{i,R})\right) + \Delta_B + \Delta_C$$
$$\geq \Delta_B + \Delta_C$$

and

$$\Delta_A \leq \Delta_B + \gamma^{T'-T}\left(\overline{h}_T(\gamma)(x_{i,B}) - \overline{h}_T(\gamma)(x_{i,R})\right)$$
$$\leq \frac{1}{3(1-\gamma)} - \Delta_C + \gamma^{T'-T}\left(\overline{h}_T(\gamma)(x_{i,B}) - \overline{h}_T(\gamma)(x_{i,L})\right)$$
$$\leq \frac{1}{3(1-\gamma)} - \Delta_C \leq \frac{1}{3(1-\gamma)}.$$

Combining the two inequalities above, we have that

$$\frac{1 - \gamma^{T'-T}}{1 - \gamma} = \Delta_A + \Delta_B + \Delta_C \leq \frac{2}{3(1 - \gamma)}.$$

This requires $\gamma^{T'-T} \geq \frac{1}{3}$, which won't be true for $T' > T + \frac{\ln 3}{\ln(1/\gamma)}$. So at least one of the first goal is achieved at time $T'$. Next we claim we can achieve one of the following goals at time $T'' \geq T'$

- $\overline{h}_{T''}(\gamma)(x_{i,L}) > \overline{h}_{T''}(\gamma)(x_{i,R}) + \frac{1}{3(1-\gamma)}$.

- $\overline{h}_{T''}(\gamma)(x_{i,R}) > \overline{h}_{T''}(\gamma)(x_{i,B})$ and $h_{T''}(x_{i,R}) = 1$.

- We make at least one mistake and the hypotheses we can eliminate is no more than the mistakes we make.

From the derivation above, we can assume the third case never happens after $T$. If the first case also never happens after $T'$, we know it must hold that $\overline{h}_t(\gamma)(x_{i,R}) > \overline{h}_t(\gamma)(x_{i,B})$ for any $t \geq T'$. So if there exists $t \geq T'$ such that $h_t(x_{i,R}) = 1$, we can finish proving the claim. On the other hand, if $h_t(x_{i,R}) = 0$ for all the $t \geq T'$, for $T'' = T' + \frac{\ln 1.5}{\ln(1/\gamma)}$, we have that

$$\overline{h}_{T''}(\gamma)(x_{i,L}) - \overline{h}_{T''}(\gamma)(x_{i,R}) \geq \sum_{t=T'}^{T''-1} \gamma^{T''-1-t} \left( h_t(x_{i,L}) - h_t(x_{i,R}) \right)$$

$$= \frac{1 - \gamma^{T''-T'}}{1 - \gamma}$$

$$\geq \frac{1}{3(1 - \gamma)}.$$

This is because $\overline{h}_{T'}(\gamma)(x_{i,L}) \geq \overline{h}_{T'}(\gamma)(x_{i,R})$ and $h_t(x_{i,L}) = 1$ for all the $t \geq T$. It means the first case will happen at $T''$ which causes contradiction.

If the second case happens, we can choose $x_{T''} = x_{i,R}$. We will make a mistake and eliminate $h_i$.

If the first case happens, we will choose $h_i$ as the target classifier. For $t = T'', \ldots, T'' + \frac{\ln(4/3)}{\ln(1/\gamma)} - 1$, it will always hold that $\overline{h}_t^{\gamma}(x_{i,L}) > \overline{h}_t^{\gamma}(x_{i,R})$ because

$$\overline{h}_t^{\gamma}(x_{i,L}) - \overline{h}_t^{\gamma}(x_{i,R})$$

$$\geq \gamma^{t-T''} \left( \overline{h}_{T''}(\gamma)(x_{i,L}) - \overline{h}_{T''}(\gamma)(x_{i,R}) \right) - \sum_{s=T''}^{t-1} \gamma^{t-1-s}$$

$$\geq \gamma^{t-T''} \frac{1}{3(1 - \gamma)} - \frac{1 - \gamma^{t-T''}}{1 - \gamma}$$

$$\geq \frac{4\gamma^{t-T''} - 3}{3(1 - \gamma)} > 0.$$

We will discuss all the cases during $t = T'', \ldots, T'' + \frac{\ln(4/3)}{\ln(1/\gamma)} - 1$ in which we can force mistakes in at least half of the rounds.

- If $\overline{h}_t^{\gamma}(x_{i,B}) > \overline{h}_t^{\gamma}(x_{i,L})$, we consider the following two cases of $h_t$.
  - If $h_t(x_{i,B}) = 1$, the adversary will choose $x_t = x_{i,L}$. The agent will manipulate to $x_{i,B}$. $h_t$ will make a mistake but learn nothing.
  - If $h_t(x_{i,B}) = 0$, the adversary will choose $x_t = x_{i,R}$. The agent will manipulate to $x_{i,B}$. $h_t$ will make a mistake but learn nothing.

- If $\overline{h}_t^{\gamma}(x_{i,B}) = \overline{h}_t^{\gamma}(x_{i,L}) > \overline{h}_t^{\gamma}(x_{i,R})$, we consider the following four cases of $h_t$.

- If $h_t(x_{i,L}) = 1$, the adversary will choose $x_t = x_{i,L}$. The agent will stay at $x_{i,L}$. $h_t$ will make a mistake.

- If $h_t(x_{i,B}) = 0$, the adversary will choose $x_t = x_{i,R}$. The agent will manipulate to $x_{i,B}$. $h_t$ will make a mistake but learn nothing.

- If $h_t(x_{i,L}) = 0$ and $h_t(x_{i,B}) = 1$, we won't force a mistake in this round. But $\overline{h}_{t+1}^{\gamma}(x_{i,B})$ and $\overline{h}_{t+1}^{\gamma}(x_{i,L})$ won't be the same and we can definitely force a mistake in round $t+1$ as discussed above.

Based on the analysis above, the learner will either make at least $\lceil \frac{\ln(4/3)}{2\ln(1/\gamma)} \rceil$ mistakes to know the target functions or make at least $|\mathcal{H}| - 1$ mistakes to eliminate all the alternative functions. It is easy to show that $\lceil \frac{\ln(4/3)}{2\ln(1/\gamma)} \rceil \geq \frac{\ln(4/3)}{4}(1-\gamma)^{-1}$ for any $0 < \gamma < 1$. So the mistake bound is $\Omega(\min\{(1-\gamma)^{-1}, |\mathcal{H}|\})$. For Littlestone dimension $d$, we can create $d$ independent copies of the above instance. The mistake bound will be $\Omega(\min\{d(1-\gamma)^{-1}, |\mathcal{H}|\})$.

$\square$

## C. Omitted Details and Proofs from Section 5

**Observation C.1** (Formal version of Observation 5.2). *There exists a hypothesis class $\mathcal{H} = \{h_1, h_2\}$ of size 2 and a manipulation graph $G$ with $k_{in}, k_{out} \leq 2$ such that for any mean-based learner $\mathcal{A}$, the decision maker will suffer at least $\sum_{t=\frac{T}{2}+1}^{T} \min\{\sigma_t^{\mathcal{A}}(\frac{t-1-T/2}{t-1}), c_t\}$ number of mistakes, where $c_t = \Omega(1)$ is the probability of choosing empirical best neighbor.*

Note that learning/exploration rate is usually set to be $\varepsilon_t = \frac{1}{\sqrt{T}}$ in Multiplicative Weights/$\varepsilon$-Greedy when the number of rounds is $T$, then we have $\sum_{t=\frac{T}{2}+1}^{T} \sigma_t^{\mathcal{A}}(\frac{t-1-T/2}{t-1}) \geq \Omega(\sqrt{T})$ for both. This implies that Multiplicative Weights/$\varepsilon$-Greedy will suffer infinite number of mistakes as $T$ goes to infinity in the strategic realizable setting.

*Proof.* Consider a manipulation graph $G$ shown in Fig 5, where there are three nodes and a hypothesis class $\mathcal{H} = \{\mathbb{1}_{x=x_L}, \mathbb{1}_{x=x_R}\}$.

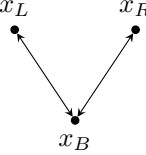

*Figure 5.*

For any time horizon $T$, in the first $T/2$ rounds, the adversary always picks agent $(x_t, y_t) = (x_B, 1)$, which is consistent with either hypothesis in $\mathcal{H}$. After the first $T/2$ rounds, we compare $\overline{h}_{T/2}(x_L)$ and $\overline{h}_{T/2}(x_R)$. We pick the target function to be $h^\star = \mathbb{1}_{x=x_R}$ if $\overline{h}_{T/2}(x_L) \geq \overline{h}_{T/2}(x_R)$, and $h^\star = \mathbb{1}_{x=x_L}$ otherwise. W.l.o.g., assume that $\overline{h}_{T/2}(x_L) \geq \overline{h}_{T/2}(x_R)$ and $h^\star = \mathbb{1}_{x=x_R}$. Then at round $t = T/2 + 1, \ldots, T$, we have $(t-1) \cdot (\overline{h}_{T/2}(x_R) - \overline{h}_{T/2}(x_L)) \leq t - 1 - T/2$.

Then at round $t = T/2 + 1, \ldots, T$, we consider the following cases:

- $\overline{h}_t(x_B) > \overline{h}_t(x_L)$ and $h_t(x_B) = 1$. In this case, the adversary will pick $(x_t, y_t) = (x_L, 0)$, then with probability at least $c_t$, the agent will manipulate to the empirically best neighbor $x_B$ and the learner makes a mistake.

- $\overline{h}_t(x_B) > \overline{h}_t(x_L)$ and $h_t(x_B) = 0$. In this case, $\overline{h}_t(x_R) - \overline{h}_t(x_B) < \overline{h}_t(x_R) - \overline{h}_t(x_L) \leq \frac{t-1-T/2}{t-1}$. Then the adversary will pick $(x_t, y_t) = (x_R, 1)$, then with probability at least $\sigma_t^{\mathcal{A}}(\frac{t-1-T/2}{t-1})$, the agent will manipulate to the empirically second best neighbor $x_B$ and the learner makes a mistake.

- $\overline{h}_t(x_B) \leq \overline{h}_t(x_L)$ and $h_t(x_L) = 0$. In this case, the adversary will pick $(x_t, y_t) = (x_B, 1)$, then with probability at least $\sigma_t^{\mathcal{A}}(\frac{t-1-T/2}{t-1})$, the agent will manipulate to $x_L$ and the learner makes a mistake.

- $\overline{h}_t(x_B) \leq \overline{h}_t(x_L)$ and $h_t(x_L) = 1$. In this case, the adversary will pick $(x_t, y_t) = (x_L, 0)$, then with probability at least $c_t$, the agent will manipulate to $x_L$ and the learner makes a mistake.

As a result, in all cases, the learner makes a mistake in each round $t > \frac{T}{2}$ with probability $\Omega(\min\{\sigma_t^{\mathcal{A}}(\frac{t-1-T/2}{t-1}), c_t\})$. Summing over $t \in [\frac{T}{2} + 1, T]$ completes the proof. $\qquad\square$

