# OpenReview forum: "Should Decision-Makers Reveal Classifiers in Online Strategic Classification?"
_ICML.cc/2025/Conference — ICML 2025 poster_

### Official Review · Reviewer_T9UQ · 2025-02-26

**Overall Recommendation:** 3

**Summary:**

This paper mainly proposes regret bounds for online strategic classification under two novel settings: (i) Revealed-Adv where the agents can still manipulate their feature arbitrarily even when the feature vector after manipulation is still negative. In this case the decision-maker makes $\Omega(k_{in})$ times more mistakes; (ii) agents do not know current policy $h_t$. Instead, they best respond to an exponential average of previous classifiers. In this case, an additional $\frac{1}{log(1/\gamma)}$ portion of mistakes are expected.

**Claims And Evidence:**

1. The contribution is very clear and the proposed settings mostly make sense in strategic classification settings;
2. The theorems are clear, and the explanations linking setting (ii) and (iii) are helpful. The intuitions introduced in Figure 1 is clear;
3. The theoretical results seem to be intuitively correct.

**Essential References Not Discussed:**

The paper already discussed related work. In (II) of Section 1.1, there are other literature quite related but not mentioned. For example, [1] studies a setting where agents arrive sequentially and also best respond to previous decision policy instead of the current one, although the authors mainly focused on welfare and fairness; [2] focused on online strategic classification in a continuous setting where the feature can be in $R^d$.

[1] Xie, T., & Zhang, X. (2024). Automating data annotation under strategic human agents: Risks and potential solutions. arXiv preprint arXiv:2405.08027.

[2] Shen, L., Ho-Nguyen, N., Giang-Tran, K. H., & Kılınç-Karzan, F. (2024). Mistake, Manipulation and Margin Guarantees in Online Strategic Classification. arXiv preprint arXiv:2403.18176.

**Experimental Designs Or Analyses:**

There is no experiment.

**Methods And Evaluation Criteria:**

This is a purely theoretical work, so there are no evaluations. For methods, Algorithm 1-2 are plausible.

**Other Comments Or Suggestions:**

I think overall the paper is well-written, and my questions are listed above.

**Other Strengths And Weaknesses:**

1. I am unsure about whether the "adversarial tie-breaking" is actually practical. In your setting, each feature vector has an $N_{out}$ and there seems to be no cost. However, in classic SC setting, manipulating the feature should incur a cost. This is to say, if there is no utility gain, the agents will never move and this makes sense.

2. The paper assumes a finite and realizable setting. Specifically, it seems that the feature vectors are countable. This may limit practical usage.

**Questions For Authors:**

N/A

**Relation To Broader Scientific Literature:**

This paper can make a meaningful contribution to the learning bounds of online strategic classification. Specifically, the idea that decision-maker does not reveal classifier and the agents respond to the previous classifiers was proposed recently and the learning bounds are worthwhile researching.

**Theoretical Claims:**

I only read through proof of Theorem 3.1 - 3.3 which mostly make sense.

---

> ### Author Rebuttal · Authors · 2025-04-01
>
> Thank you for your thoughtful comments. We address them below.
>
> > adversarial tie-breaking
>
> We appreciate the reviewer’s concern and want to clarify that we do *not* intend to make adversarial tie-breaking a realistic behavioral assumption. Instead, we use it as a conceptual and technical intermediate step toward analyzing the more realistic setting (\gamma-Weighted) where agents *cannot* observe the currently implemented classifier and instead best respond to an *estimate* of it. As we discussed in Lines 235-251 of our manuscript, in such settings, estimation error can cause the agents to perceive one node as slightly better than the other node, even if the true classifier might assign them the same label. For example, if $\tilde h(v) = h(v)$ for all $v \in N_{out}(x)$, but $\tilde h(x) = h(x) - \epsilon$, the agent would not remain at $x$, behaving as if they are breaking ties adversarially.
> The purpose of Theorem 3 is precisely to highlight the mistake bound gap between the case where agents only perceive a perturbed version of the true hypothesis $h$ versus the case where they perceive $h$ perfectly. We treat adversarial tie-breaking as a proxy for incorporating such estimation errors into the mistake bound analysis, which serves as a building block towards the gamma-Weighted setting.
> Second, our upper bound in Theorem 4.1 holds for both tie-breaking ways. We note that our lower bound in Theorem 4.4 consists of two parts: $\min(d/\log(1/\gamma), |H|)$ and $d \cdot k_{\text{in}} \cdot k_{\text{out}}$* , where the first part still holds for the standard tie-breaking case with a slight refinement of the proof. The second part will be reduced to $d\cdot k_{\text{out}}$ in the standard tie-breaking case. The dependency on $k_{\text{in}}$ in unrevealed classifier + standard tie-breaking is left as an open problem.
>
> *There is a typo in the original statement of Theorem 4.4, $ k_{\text{in}}$ should be replaced by $k_{\text{in}} \cdot k_{\text{out}}$.
>
> > finite feature space
>
> First, we would like to clarify that our model does not require the feature or action space to be finite: one can define a generic manipulation graph structure over any (potentially infinite or continuous) feature space by treating features as vertices and defining edges as possible manipulations. However, the *learnability* of an instance — in both our work and prior works [Ahmadi et al, 2023, 2024; Cohen et al, 2024] — relies on the finiteness of the *maximum degree* of the manipulation graph.  Without the finite degree assumption, learning in infinite graphs is generally intractable. This has also been noticed by prior work eg [Shao et al, NeurIPS 2023, “Strategic Classification under Unknown Personalized Manipulation”], which shows that when the feature space is continuous and agents can manipulate to any point within a ball around the initial features, mistake bounds are usually $\Omega(|H|)$.
>
> Second, we would like to remark that when the feature space is continuous but satisfies certain smoothness conditions, it is often possible to discretize the space and then construct a finite manipulation graph on a covering net. Our results then apply to this discretized finite graph.
>
>
> > realizable setting
>
> Our goal is to study the gap between the setting where the currently implemented hypothesis is revealed and the setting where it is not. The agnostic case remains open in the literature even in the revealed classifiers setting —as also noted by reviewer MqF7. However, we believe that similar ideas as used in Algorithms 1 and 3 can be applied to the agnostic setting too.

---

### Official Review · Reviewer_MqF7 · 2025-03-09

**Overall Recommendation:** 4

**Summary:**

This work considers a strategic online classification setting. At each round $t$, the learner selects classifier $h_t : \mathcal{X} \in \{0,1\}$ and an agent with true feature $x_t \in \mathcal{X}$ and label $y_t \in \{0,1\}$ selects a manipulated feature $v_t \in \mathcal{X}$. After the round, $v_t$ and $y_t$ are revealed to the learner. This work supposes that the agent selects $v_t$ among the out-neighbors of $x_t$ in a directed graph, in order to maximize $\tilde{h}_t(v_t)$, where $\tilde{h}_t : \mathcal{X} \to [0,1]$ is a guess for $h_t$. Previous work examined this problem thoroughly when $\tilde{h}_t = h_t$ and the agent breaks ties favorably. This work considers the setting where $h_t$ is not revealed until after round $t$, and the learner uses a weighted sum of previous classifiers, according to a discount factor $\gamma$. Under a standard realizability assumption, they design new algorithms and upper bound the number of mistakes made by the learner, accompanied with constructions lower bounding the number of mistakes by any learning algorithm. When $\tilde{h}_t = h_t$ but tie-breaking is adversarial, they show that their is already a significant overhead in the mistake bound based on the in-degree of the manipulation graph. In the weighted sum case, there is further overhead according to the agents' time horizon $1/\log(1/\gamma)$. This provides a simple strategic learning setting where withholding access to the classifier degrades performance.

## Update after Rebuttal
I have increased my score to accept based on their clarifications and promised corrections.

**Claims And Evidence:**

Yes, this is a theoretical paper and (almost) all of the results have proofs which appear sound. The only piece I slightly struggled with was the proof of Theorem 4.4. There is one piece of their claim which I don't think is properly treated by the proof and one typo in the statement (I think). For the rest of the proof, I can follow the line-by-line arguments but I think a better proof overview would be helpful, since this result is key to the paper's thesis. The bound is somewhat natural, since $1/\log(1/\gamma)$ is roughly the effective memory length of the agents, so I would be surprised if there was any major issue.

I feel confident vouching for correctness of the other results, whose proofs I found pretty clear.

I am willing to raise my score if the authors address my concerns with Theorem 4.4.

**Essential References Not Discussed:**

Nothing comes to mind.

**Experimental Designs Or Analyses:**

n/a

**Methods And Evaluation Criteria:**

n/a

**Other Comments Or Suggestions:**

For me, it feels a bit more intuitive to imagine the agent maintaining a distribution over classifiers and manipulating their feature to maximize the probability of a positive label. This of course coincides with taking a weighted sum over classifiers though, so there is no issue.

I didn't understand the phrase "With probability at least 1/3, the agent will stay at xi,B because it is one of the best candidates." on page 15. Isn't the agent's choice up to us in this LB construction? I.e. we can have this occur all of the time. In any case, I don't think this point affects the result, but I want to make sure I'm not misunderstanding something.

This is a small nit, but I prefer using $(1-\gamma)^{-1}$ instead of $1 + 1/\log(1/\gamma)$. These are equivalent up to constant factors and the former is commonly used for effective time horizons in RL. I think it's worth noting somewhere.

Double parentheses in last citation on page 2

**Other Strengths And Weaknesses:**

I actually think the technical contributions with respect to tie-breaking are more interesting than that of revealing the classifier, which the paper is framed around. In particular, the $k_\mathrm{in}$ dependence seems to be due to the tie-breaking choice rather than not revealing the classifier. The dependence on $1/\log(1/\gamma)$ is much less surprising to me.

The realizability assumption is rather strong, and these results would definitely be more compelling without it. This is an open problem even when classifiers are revealed immediately though, so I don't hold that against the authors.

**Questions For Authors:**

What does this setting look like if the agents arrive stochastically? Can the Littlestone dimension can be reduced to a VC dimension?

Algorithm 1 seems pretty generic and I could imagine the same idea being useful in other settings. Have you thought about any other applications / has any similar idea appeared in the literature?

**Relation To Broader Scientific Literature:**

There is a fair amount of literature on strategic classification, which is discussed in the related work. Their results shine some light on the importance of tie-breaking assumptions made in previous works for the online setting where $\tilde{h}_t = h_t$. The extension they consider to the weighted average case is a pretty natural one. I don't think it is that surprising that their agents with memory are harder to learn against, but it is good to establish.

**Theoretical Claims:**

Yes, I read through all of the proofs, as mentioned above. For Theorem 4.4, I had two comments. First, shouldn't $k_\mathrm{out}$ appear in the LB? I think this is just a typo. Second, I don't see why taking $\gamma \to 0$ is okay for the first part of the proof. The theorem was claimed to hold for each fixed $\gamma$ and it seems non-trivial to extend the LB in Theorem 3.3 when $\gamma$ is bounded away from 0.

---

> ### Author Rebuttal · Authors · 2025-04-01
>
> We would like to thank the reviewer for the thoughtful comments. We address them below.
> > lower bound statement in Theorem 4.4
>
> Thanks for catching the issue in the theorem statement. The lower bound in Theorem 4.4 consists of two parts: $\min( d/\log(1/\gamma), |H|)$ and $d \cdot k_{\text{in}} \cdot k_{\text{out}}$. The first bound holds for all $\gamma\in(0,1)$, and its proof is provided in Appendix B.3. The second bound emphasizes the dependency on maximum in-degree and is established only when $\gamma\to0$. Its proof follows from a modified version of that of Theorem 3.3.
>
> We agree that the theorem statement should be more precise, and will update in revisions to clarify that the second bound is only established in the $\gamma\to0$ regime. In the following, we provide more details for this modified proof and will add them to future versions of this paper.
>
> Consider the graph that is the same as that in Figure 2 but all nodes in the first layer (namely $x_1,x_2,\ldots,x_{k_1}$) are connected by a clique. This modification gives $k_{\text{out}}=k_1+k_2$ and $k_{\text{in}}=k_2$.
>
> For $\gamma\to0$, best responding to $\tilde{h}^\gamma_t$ is equivalent to best responding to $h_{t-1}$. Now, we perform a case discussion of both $h_{t-1}$ — which the agent $x_t$ best responds to — and the current classifier $h_t$. We will construct an adversary that induces at least one mistake in every two rounds in the first $2k_1k_2$ rounds. Now we assume that the learner does not make a mistake at round $t-1$ and show how to induce a mistake in round $t$.
>
> * Case 1: If $h_{t-1}$ labels $x_0$ by positive. Then if $h_t(x_0)=1$, the adversary picks $(x_t,y_t)=(x_0,0)$ and induces a false positive mistake. If $h_t(x_0)=0$, then the adversary picks $(x_t,y_t)=(x_{i^*},1)$, which manipulates to $x_0$ and induces a false negative mistake while revealing no information. These correspond to cases 1 and 2 of the proof of Theorem 3.3.
>
> * Case 2: If $h_t$ labels any leaf node $x_{i,j}$ as positive, then the adversary picks $(x_t,y_t)=(x_{i,j},0)$. This induces a false positive mistake and removes one hypothesis. Since the learner made no mistake in round $t-1$, we can assume that $h_{t-1}$ labels all leaf nodes as negative. This case corresponds to case 4 of Theorem 3.3.
>
> * Case 3: We are left with the case where $h_{t-1}$ labels $x_0$ and all leaf nodes as negative. Due to adversarial tie-breaking and the added clique, we can assume that all nodes in $\{x_0,x_1,\ldots,x_{k_1}\}$ manipulates to the same node $x_i$ in response to $h_{t-1}$. If $h_t(x_i)=1$, then the adversary can choose $(x_t,y_t)=(x_0,0)$ and induce a false positive mistake. If $h_t(x_i)=0$, then the adversary chooses $(x_t,y_t)=(x_{i^*},1)$ and induces a false negative mistake while revealing no information about $i^*$. This case is a modification of case 3 of Theorem 3.3 that accounts for historical best response.
>
> We will correct this in the paper.
>
> > the phrase "With probability at least 1/3”
>
> We allow agents to best respond to the weighted average classifier with adversarially tie-breaking in this setting and we focus on adversarially choosing the specific $x_t$ at each time. Instead of finding the worst instance among all possible cases, it is enough to show our desired instance can happen with $\Theta(1)$ probability.
>
> > agents arrive stochastically
>
> Since we focus on mistake bounds in the realizable setting, they cannot be reduced to the VC dimension in the stochastic setting, even in standard online learning without strategic behavior. However, for regret bounds in the agnostic setting, such a reduction is possible. We believe it would be interesting to explore regret bounds in the agnostic setting, both when the classifier is revealed and when it is not.
>
> > application of Algorithm 1
>
> Algorithm 1 has two main components: the weighted voting of experts and the choice of threshold/updating mechanism of experts. The high-level idea of weighted expert voting is fundamental and has been widely applied in many settings. Choosing a biased threshold that favors false positives is useful in scenarios where false positives and false negatives carry different amounts of information about the true hypothesis.
>
> > Novelty of our results
>
> We would like to re-emphasize the technical novelty of our results.
> First, as discussed in our response to reviewers WPjm and T9UQ regarding adversarial tie-breaking, we show that when agents only observe a perturbed version of the current predictor $h_t$—rather than the exact $h_t$—the mistake bound increases by a factor of $k_\text{in}$ (see Corollary 3.2 and Theorem 3.3).
>
> Second, in the weighted sum of history case, as the reviewer pointed out, $1/\log(1/\gamma)$ can be seen as the effective memory length of the agent. One interesting implication of this lower bound is that, in the worst case, the agent may need to “forget” its entire history—incurring $\Omega(1/\log(1/\gamma))$ mistakes—before it can make correct decisions again.

---

> > ### Comment · Reviewer_MqF7 · 2025-04-03
> >
> > Thank you for the thorough response - I am raising my score to accept.

---

> > > ### Author Response · Authors · 2025-04-03
> > >
> > > Thank you for raising the score! We will update the manuscript accordingly.

---

### Official Review · Reviewer_q9sy · 2025-03-14

**Overall Recommendation:** 4

**Summary:**

This paper studies how hiding the classifier from the agents affects the performance of strategic classification. The result shows that hiding the classifier from the agents significantly increase the number of mistakes the decision maker would make, in proportion to the maximum in-degree of the manipulation graph. The paper proposed two algorithms for strategic learning with adversarial and $\gamma$-weighted agents and calculated the upper bound on the number of mistakes each algorithm makes. The authors also provided special examples where the upper bound is attained.

**Claims And Evidence:**

The central claim is hiding classifiers would increase the number of mistakes of the decision maker. This is well supported by the reduction from the $\gamma$-weighted version to the adversarial agents. Every claims are soundly supported by theoretical construction and proofs.

**Essential References Not Discussed:**

Not recognized.

**Experimental Designs Or Analyses:**

The paper is purely theoretical, relying on algorithms and constructed examples (e.g., Figure 1 and 2).

**Methods And Evaluation Criteria:**

The key methodology is to reduce the target situation ($\gamma$-weighted case) to the adversarial setting and derived bounds of the latter based on Littlestone dimensions. The method is innovative and appropriate. The evaluatoin metric is the number of mistakes the decision maker made throughout the interaction process. The evaluatoin metrics is reasonable and has well-documented benchmarks.

**Other Comments Or Suggestions:**

No.

**Other Strengths And Weaknesses:**

The most important innovation of this paper lies in the idea of connecting the target setting ($\gamma$-weighted) agents to the adversarial agents. The paper contributes to the strategic classification community by theoretically supporting the commonly adopted assumption that hiding classifiers to the agents does not benefits the decision maker.

The weakness could be in the limitations of the action space where in usual cases agents may have access to an infinite or even continuous action space.

**Questions For Authors:**

No.

**Relation To Broader Scientific Literature:**

The work extends the strategic classification framework to non-transparent settings. While it’s commonly assumed and argued that revealing classifiers to agents is reasonable and practical, rigorous theoretical studies on its benefits are scarce. This work fills this gap in the literature.

The model introduced in this work, which incorporates discounted agent memory and adversarial tie-breaking, is novel in my understanding.

Section 5 connects the work to repeated Stackelberg games with learning agents. The authors thoroughly discuss the challenges of adapting the current model to the learning agent problem and provide insights into the implications of the findings in this context.

**Theoretical Claims:**

I checked all theorems and lemma except Observation 5.2. The theoretical results are correct as far as I can tell.

---

> ### Author Rebuttal · Authors · 2025-04-01
>
> Thank you for the thoughtful comments and for examining our proofs. We address them below.
>
> > infinite/continuous feature space
>
> First, we would like to clarify that our model does not require the feature or action space to be finite: one can define a generic manipulation graph structure over any (potentially infinite or continuous) feature space by treating features as vertices and defining edges as possible manipulations. However, the *learnability* of an instance — in both our work and prior works [Ahmadi et al, 2023, 2024; Cohen et al, 2024] — relies on the finiteness of the *maximum degree* of the manipulation graph.  Without the finite degree assumption, learning in infinite graphs is generally intractable. This has also been noticed by prior work eg [Shao et al, NeurIPS 2023, “Strategic Classification under Unknown Personalized Manipulation”], which shows that when the feature space is continuous and agents can manipulate to any point within a ball around the initial features, mistake bounds are usually $\Omega(|H|)$.
>
> Second, we would like to remark that when the feature space is continuous but satisfies certain smoothness conditions, it is often possible to discretize the space and then construct a finite manipulation graph on a covering net. Our results then apply to this discretized finite graph.

---

### Official Review · Reviewer_WPjm · 2025-03-14

**Overall Recommendation:** 3

**Summary:**

This work studies an online classification setting, where arriving points (agents) can manipulate their features according to a manipulation graph, where the rationale to to obtain more favorable predictions. Under a realizability assumption, the authors study a scenario where the decision maker does not reveal its currently deployed predictor to agents, who instead manipulate their features as a response to the weighted average of past predictors in the interaction. Under an additional assumption of adversarial manipulations by agents, the authors prove upper and lower mistake bounds for the problem, and show a gap for the full information analogue (though without allowing manipulations to be adversarial), studied in previous work. The paper concludes with examining situations where agents run no regret algorithms instead of best responding, and demonstrate the difficulty of learnability in their model.

**Claims And Evidence:**

The paper is fully theoretical. I did not read the appendix with the proofs, but from glancing, it seems like the resulted are well established.

**Essential References Not Discussed:**

In the context of revealing predictors to agents, one line of work that I think is relevant to mention is on providing explanations to agents instead of revealing the predictor (e.g. Counterfactual Explanations, Tsirtsis and Rodriguez 2020), where the agents are given a suggested point to manipulate to, and the prediction at that new point. It is important to point out that there are alternative actionable signals that can be provided to agents to induce strategic modification, and the choice is not only between revealing a predictor or obscuring it (and even if a predictor is revealed and is of high complexity, a common agent may find it difficult to even calculate their best response). I believe the authors should include such a discussion in relevant previous work.

**Experimental Designs Or Analyses:**

N/A.

**Methods And Evaluation Criteria:**

N/A

**Other Comments Or Suggestions:**

Algorithm 3 should appear in the main text. I understand that there are space limitations, however it is central to this work and should be included.

**Other Strengths And Weaknesses:**

Strengths:
1. The paper is very well written, I enjoyed reading it.

2. The approaches for solving the main problem in this work are original, creative, and I found the reduction approach in the paper interesting.

3. The problem of whether classifiers should be revealed in strategic settings is central, and clearly motivated.



Weakness:
I am slightly concerned regarding the significance of obtained results, due to the following points:

1. In the scenario of the paper, there is strong reason to believe that a decision maker who does not want to reveal their current model, would not want to reveal past deployed policies (e.g. there might be many similarities, and the developed predictors may be proprietary and allow the decision maker to maintain competitive edge).

2. The scenario of adversarial manipulation, in which agents may still manipulate to different points even if they do not benefit from such action seems to be rather unnatural (especially that usually such manipulations have cost/effort for agents). To my understanding, the lower bounds in Theorems 3.3, 4.4 strongly rely on this assumption.

3. The realizability assumption in the basis of this work (while maybe necessary to allow for the analysis) is rather restrictive.

4. There are alternatives to revealing the model in inducing agent responses, which may aid to circumvent the tensions between the two extremes (see more in essential references section).

**Questions For Authors:**

1. Why did you opt to study the problem under the additional (and largely unnatural in a strategic agents context) assumption of adversarial manipulations? Have you considered the non-adversarial + unrevealed classifier case?

2. In line 340, you state "a false negative mistake can occur only if a false positive mistake happened in the previous round". Can you explain why this is the case?

3. Regarding Algorithm 3 (described in the paragraph starting in line 361) --- why can't there be instability in the classification where there are recurring mistakes but still inability to make progress according to the described scheme? The argument is probably formalized in the appendix, but I am wondering about the main ideas in the proof.

**Relation To Broader Scientific Literature:**

The paper contributes to the line of work on strategic classification, and particularly to studying the question regarding the implications of not revealing the deployed predictors to agents. The paper follows a line of work modelling possible manipulations as graphs.

**Theoretical Claims:**

I did not.

---

> ### Author Rebuttal · Authors · 2025-04-01
>
> Thank you for the thoughtful comments. We address them below.
>
> > revealing the past deployed policies
>
> We agree that the decision-maker might not want to reveal their past models explicitly. However, historical outcomes often reveal past classifiers implicitly. For example, we can observe which students were admitted by a college in previous years (based on SAT scores, GPA, extracurriculars) or which loan applicants were approved—and use this information to estimate the predictors that were likely implemented in the past. In some specific cases, the model is effectively made public: in college admissions in countries with unified entrance exams, universities often reveal the threshold classifier by publishing the minimum admission scores after each admissions cycle.
>
> > adversarial tie-breaking
>
> We appreciate the reviewer’s concern and want to clarify that we do *not* intend to make adversarial tie-breaking a realistic behavioral assumption. Instead, we use it as a conceptual and technical intermediate step toward analyzing the more realistic setting (gamma-Weighted) where agents *cannot* observe the currently implemented classifier and instead best respond to an *estimate* of it. As we discussed in Lines 235-251 of our manuscript, in such settings, estimation error can cause the agents to perceive one node as slightly better than the other node, even if the true classifier might assign them the same label. For example, if $\tilde h(v) = h(v)$ for all $v \in N_{out}(x)$, but $\tilde h(x) = h(x) - \epsilon$, the agent would not remain at $x$, behaving as if they are breaking ties adversarially.
>
> The purpose of Theorem 3 is precisely to highlight the mistake bound gap between the case where agents only perceive a perturbed version of the true hypothesis $h$ versus the case where they perceive $h$ perfectly. We treat adversarial tie-breaking as a proxy for incorporating such estimation errors into the mistake bound analysis, which serves as a building block towards the gamma-Weighted setting.
> Second, our upper bound in Theorem 4.1 holds for both tie-breaking ways. We note that our lower bound in Theorem 4.4 consists of two parts: $\min(d/\log(1/\gamma), |H|)$ and $d \cdot k_{in} \cdot k_{out}$* , where the first part still holds for the standard tie-breaking case with a slight refinement of the proof. The second part will be reduced to $d\cdot k_{out}$ in the standard tie-breaking case. The dependency on $k_{\text{in}}$ in unrevealed classifier + standard tie-breaking is left as an open problem.
>
> *There is a typo in the original statement of Theorem 4.4, $ k_{in}$ should be replaced by $k_{in} \cdot k_{out}$.
>
> > realizability assumption
>
> Please see the response to reviewer T9UQ.
>
> > alternatives to revealing the model in inducing agent responses
>
> We completely agree that there are interesting settings between the two extremes, and we see it as an exciting open direction to study the spectrum of information that may be revealed. We will add [Tsirtsis and Rodriguez 2020] to the related work section and include a discussion along these lines.
>
> > explanation for line 340
>
> The complete proof for Lemma 4.2 is in Appendix B.2, and we’d like to provide more explanation here. At first sight, it seems that there won’t be false negative mistakes because our algorithm is very conservative. Every positive sample $x$ will always have a neighbor $u$ that is labeled as positive by $h^\star$ and $h_t$. Such $u$ is always in $BR_{\tilde{h}_t^\gamma}(x)$ and we won’t make a false negative mistake on $x$ if $x$ manipulates to this $u$. However, we argue there still can be false negative errors for one very specific scenario. That is, $x$ manipulates to another neighbor $v$ in $BR\_{\tilde{h}\_t^\gamma}(x)$ but $h\_t(v)=0$. For such a case to happen, $\tilde{h}\_t^\gamma(v)$ must be 1 to ensure that $x$ will choose to manipulate to it with non-negative probability. That means there is a $h’$ that is kept in $E$ until time $t-1$ such that $h’(v)=1$. However, $h_t(v)=0$ means that this $h’$ is not in $E$ at time $t$ so it must be removed at time $t-1$. Since we only remove a classifier when a false positive mistake occurs, the number of false negative errors is no larger than the number of false positive errors.
>
> > main idea of Algorithm 3
>
> Algorithm 3 is a reduction to the Revealed-Adv setting, so its progress is tightly coupled with the progress of the algorithm used in the Revealed-Adv setting (Let A’ be such an algorithm, e.g., Algorithm 1). Specifically, Algorithm 3 calls A’ once every $\Phi$ mistakes it makes, so every $\Phi$ mistakes of Algorithm 3 corresponds to a single mistake made by A’. Therefore, if algorithm 3 makes recurring mistakes, then A’ must also be making recurring mistakes in the Revealed-Adv setting. However, this contradicts with the mistake bound for A’ (e.g., Corollary 3.2). Therefore, such “instability” cannot persist, and Algorithm 3 ends up having a finite mistake bound.

---

### Decision · Program_Chairs · 2025-05-01

**Decision:**

Accept (poster)

**Comment:**

This paper studies online strategic classification where the learned model is not always revealed to users. The paper establishes interesting theoretical claims regarding outcomes for the decision-maker in terms of the benefits and losses of a transparent policy. Reviewers found the writing to be clear, the contributions clearly stated, and the results to be sound and easy to follow. They also believe that the paper fills a gap in the online setting of strategic classification, and does this well. Some reviewers pointed out the strong assumptions made (e.g., realizability, action space). Several reviewers thought that the paper would benefit from better explaining the rationale for tie-breaking, and even making this point more emphasized in the paper. The authors are encouraged to take these and all other questions and concerns raised by the reviewers and implement them in the next revision.